# Centrosome Formation in the Bovine Early Embryo

**DOI:** 10.3390/cells12091335

**Published:** 2023-05-07

**Authors:** Rustem Uzbekov, Galina N. Singina, Ekaterina N. Shedova, Charles Banliat, Tomer Avidor-Reiss, Svetlana Uzbekova

**Affiliations:** 1Laboratory of Cell Biology and Electron Microscopy, Faculty of Medicine, University of Tours, 37032 Tours, France; 2Faculty of Bioengineering and Bioinformatics, Moscow State University, 119992 Moscow, Russia; 3Laboratory of Experimental Embryology, L.K. Ernst Federal Research Center for Animal Husbandry, Moscow Region, 142132 Podolsk, Russia; 4Ecole Supérieure d’agricultures (ESA), Unité de Recherche sur les Systèmes D’élevage (URSE), 55 rue Rabelais BP, 30748 Angers, France; 5Department of Biological Sciences, University of Toledo, Toledo, OH 43606, USA; 6UMR Physiologie de la Reproduction et des Comportements (PRC), INRAE, CNRS, Université de Tours, IFCE, 37380 Nouzilly, France

**Keywords:** centriole, centrosome, zygote, early development, cleavage divisions, polar corpuscles

## Abstract

Centrosome formation during early development in mice and rats occurs due to the appearance of centrioles de novo. In contrast, in humans and other non-rodent mammals, centrioles are thought to be derived from spermatozoa. Ultrastructural study of zygotes and early embryos of cattle at full series of ultrathin sections show that the proximal centriole of the spermatozoon disappears by the end of the first cleavage division. Centrioles appear in two to four cell embryos in fertilized oocytes and in parthenogenetic embryos. Centriole formation includes the appearance of atypical centrioles with randomly arranged triplets and centrioles with microtubule triplets of various lengths. After the third cleavage, four centriolar cylinders appear for the first time in the blastomeres while each embryo still has two atypical centrioles. Our results showed that the mechanisms of centriole formation in different groups of mammals are universal, differing only in the stage of development in which they occur.

## 1. Introduction

Despite the great importance of centrioles in various processes of cell functioning, their role remains only partially understood. One of the remaining questions concerns centriolar inheritance during early embryonic development. The appearance of centrioles has been described in various species, demonstrating that the process can occur with noticeable differences between evolutionarily close species and more similarly in evolutionarily distant animals.

The question of the role of centrioles in early development interested biologists by the end of the 19th century, after the discovery of dense granules at the spindle poles, to which the chromosomes moved during anaphase of mitosis [1]. It was also found that during fertilization, such granules were introduced into the zygote by spermatozoa. If the oocyte was fertilized by two spermatozoa, formation of tripolar mitosis occurred. Based on these observations, Boveri formulated the hypothesis of the paternal inheritance of centrioles [2]. This hypothesis has long been dominant, especially when Mazia and colleagues showed in their elegant experiments the role of the sperm centrosome during the first embryonic divisions in sea urchins [3,4,5]. In sea urchins, two centrioles were preserved in spermatozoa, both of which were found in the zygote [6]. Though it seemed that the question of centriole inheritance was resolved, subsequent research questioned the universality of this beautiful hypothesis.

In all animals, centrioles disappear during oogenesis [7,8,9,10]. In mice, the distal and proximal centrioles degrade at the final stage of sperm maturation in the epididymis [11]; therefore, after fertilization, mouse zygotes do not possess centrioles of either paternal or maternal origin.

Using transmission electron microscopy (TEM), it was found in mice, rats, and other rodents that the first embryonic cleavage occurs in the complete absence of typical centrioles in embryonic blastomeres [12]. The appearance of centrioles was evidenced from the 8- up to even the 32-cell stage of mouse embryos [7,13,14,15,16], and was preceded by embryo genome activation, which occurs at the 2-cell stage [17] (Flach et al., 1982). Typical centrioles appear asynchronously in various cell types—blastocyst trophectoderm and the inner cell mass [15]—where they later start forming cilia [18]. The early embryo contains both the proteins accumulated during oogenesis and newly synthesized proteins, which can construct de novo centrioles in mouse blastomeres. Regulatory proteins of embryonic origin are likely required, since otherwise, the appearance of the centrioles would begin at an earlier stage. Maternal proteins likely remain in the ooplasm after maternal centriole disassembly during oogenesis and play a structural role.

Additionally, it was shown that centrioles could be formed de novo in parthenogenetically activated eggs of the sea urchin [19,20,21]. In sheep parthenogenetic embryos, no centrioles were found at the first division [22]. Parthenogenetic cattle embryos could form the first spindle and two blastomeres without the participation of sperm material; their spindle was barrel-shaped and anastral [23]. A more comprehensive study was reported on parthenogenetic development of rabbit embryos, where centrioles appeared de novo and were found only at the blastocyst stage [24].

The universality of the Boveri theory of exclusive paternal inheritance of centrioles in all organisms was experimentally refuted, as a parallel study of early mouse and sea urchin embryos showed fundamental differences in the inheritance of centrioles in these organisms [25]; thus, a more nuanced model should be proposed.

During the spermatid stage of spermiogenesis, the distal centriole gives rise to the sperm flagellum, and the proximal centriole to the centriolar adjunct [26]. Unlike in mice and other rodents, in humans and cattle the proximal sperm centriole retains its structure and penetrates into the zygote during fertilization.

The most logical and simplest explanation of the fate of centrioles in early embryo development is that the proximal and distal centrioles duplicate and thus form two diplosomes (consisting of two connected centrioles), which are located at the poles during the first division. This scheme was constructed by analogy with a detailed investigation in a sea urchin model [27]. In sea urchins, the spermatozoon introduces two functionally active centrioles into the zygote [6]. In contrast, in all studied mammalian spermatozoa, the distal daughter centriole, which gives rise to the sperm flagellum [28], completely loses its morphology [26,29,30,31,32]. Detailed analysis on ultrathin serial sections perpendicular to the axis of the flagellum showed that in primate and human spermatozoa, the two central microtubules (MT) of the axoneme continue up to the surface of the proximal centriole, and there are no centriolar MT triplets in the expected distal centriole region [31]. Moreover, the doublets of the MT flagellum diverge from each other and form a peculiar funnel expanding towards the nucleus.

Since only one intact sperm centriole enters the zygote, and diplosomes, consisting of two centrioles, are usually located at the spindle poles of somatic cells, **the first model** assumes that the proximal centriole undergoes two duplication cycles before the first division (Figure 21 in [33]). However, the occurrence of two duplication cycles of the proximal centriole has not been confirmed experimentally.

The question of the centriole’s role in spindle organization was raised again in the study of early mammalian, non-rodent embryos, particularly in humans [33,34] and sheep [22,35]. In these studies, the centrioles were observed in blastomeres after the first and second cleavages. Additionally, during the first cleavage, typical centrioles were not always found at the spindle poles, and when they were found, they were not classical diplosomes but single centrioles. To date, no studies on the first cleavages of the mammalian embryo have reported two typical centrioles in either pole.

The question remains how a single proximal centriole from the spermatozoon in non-rodent mammals can, before mitosis, form four centrioles, which are present in all dividing somatic cells. Two other models have been proposed to address this question.

**The second model** proposes that the distal centriole of the sperm restores its structure and, together with the proximal centriole, undergoes one cycle of duplication before the first cleavage (Figure 2 in [36]).

**The third model [32]** is based on localization of centrosomal and centriolar proteins to two foci in the zygote, as observed by immunocytochemistry. This model suggested the passage of one cycle of centriole duplication, though occurring on two different types of matrices: one of the newly formed centrioles was formed on the proximal sperm centriole, while the second originated from an “atypical centriole”, which is a remodeled form of the distal centriole located near the apical end of the sperm flagellum.

In the eight publications from the Sathananthan laboratory (see summary in Table 1 in [33]), only three normal human embryos (egg fertilized by one spermatozoon) were studied at the pronuclei stage. A centriole was found in only one cell; in the other two syngamy-stage embryos (metaphase of the first division), the centriole was also found in only one cell. Of the two two-cell embryos, one centriole was detected in only one embryo, while in other cells, no centrioles were found [33]. The authors noted that the irregular presence of centrioles in early embryos could be due to four different reasons: (i) centrioles are minute objects that could easily go undetected, even by TEM; (ii) serial sections may be lost during microtomy, or sections may rest on grid bars, obscuring spindle poles; (iii) centrioles may be located in thick survey sections, where they cannot be detected by light microscopy; (iv) it is difficult to orient and section spindles in a desired plane, as spindles are not visible in whole embryos at syngamy when viewed by light microscopy [34].

Altogether, the origin of the early embryonic centrioles is unclear. To resolve this gap of knowledge, our study objective was to follow the appearance of the centriole during the first cleavages in bovine embryos and determine their structure using electron microscopy. To limit the loss of microscopic data usually associated with this type of study, we have paid particular attention to several specific points of the workflow. First, we carefully studied serial sections at high magnification throughout the entire volume of zygote or embryo to prevent overlooking the centrioles. Second, we have cut the entire embryo into ultrathin serial sections, as opposed to the semi-thin or thick sections that were previously selected for observation with light microscopy. Third, we have only used one-slot grids covered with a Formvar film to prevent the loss of centrioles due to the metal jumpers of the grid for TEM.

## 2. Materials and Methods

### 2.1. In Vitro Embryo Production by Fertilization

Bovine ovaries were collected from a local slaughterhouse and transported at 36 °C to our laboratory. Cumulus–oocyte complexes (COCs) were recovered using HEPES-buffered TCM199 supplemented with 0.4 g/L bovine serum albumin (BSA) and 0.25% gentamicin. Groups of 30–60 COCs were matured in TCM199 supplemented with 10 ng/mL EGF, 19 ng/mL IGF-1, 2.2 ng/mL FGF, 5 UI/mL hCG, 10 UI/mL PMSG, 4 µg/mL transferrin, 4 µg/mL insulin, 5 ng/mL sodium selenite, 1% PG-600, 90 µg/mL L-cysteine, 0.1 mM beta-mercaptoethanol, 75 µg/mL ascorbic acid, 720 µg/mL glycine, 0.1 mg/mL glutamine, and 110 µg/mL pyruvate at 38.8 °C (5% CO_2_) for 22 h. After maturation, COCs were transferred into 250 µL of fertilization medium (Tyrode medium supplemented with 25 mM bicarbonate, 10 mM lactate, 1 mM pyruvate, 6 mg/mL fatty acid-free BSA, 10 µg/mL heparin, and 40 µg/mL gentamycin). Motile spermatozoa were recovered by Percoll washing from one Normande bull (Evolution, Noyal-sur-Vilaine, France), added to the COCs in the fertilization medium (Day 0) at a final concentration of 10^6^ spermatozoa/mL, and incubated for 22 h. On Day 1, presumptive zygotes were transferred to 500 µL of synthetic oviductal fluid (SOF) medium [37], supplemented with 0.01% of polyvinyl alcohol (SOF–PVA) without any serum or protein supplementation, and incubated at 38.8 °C with 5% CO_2_ and 5% O_2_.

### 2.2. Parthenogenetic Development of Embryos In Vitro

Bovine ovaries were purchased from a commercial slaughterhouse and transported in saline buffer containing 100 U/mL penicillin and 50 µg/mL streptomycin at 30–35 °C.

COCs were recovered by slicing the ovaries and were washed three times with TCM199 supplemented with 5% fetal calf serum (HyClone, Logan, UT, USA), 0.5 mM sodium pyruvate, 10 µg/mL heparin, and 50 µg/mL gentamycin. Only high-quality COCs were selected, i.e., those that contained oocytes with homogenous cytoplasm and several layers of cumulus cells (CC), as determined using an SMZ inverted microscope (Nikon, Tokyo, Japan) at 37 °C, as described in [38]. In vitro maturation (IVM) was performed on groups of 30–35 COCs in TCM199 culture medium supplemented with 10% fetal calf serum (HyClone, USA), 0.5 mM sodium pyruvate, 10 µg/mL FSH, 10 µg/mL LH, and 50 µg/mL gentamycin for 24 h at 38.8 °C in a humidified atmosphere with 5% CO_2_ in 95% air. After IVM, the COCs were incubated in 0.1% hyaluronidase in TCM199 supplemented with 10% FCS, 0.5 mM pyruvate, and 50 µg/mL gentamycin at 37 °C for 1 min, after which the oocytes were stripped off CC by repetitive aspirating–ejecting movements through a pipette 130 µm in diameter. Fully mature oocytes with extruded polar bodies were selected.

Mature oocytes were activated by ionomycin (5 mM) in HEPES-buffer Tyrode solution containing 114 mM NaCl, 3.1 mM KCl, 0.4 mM NaH_2_PO_4_ × 2H_2_O, 2 mM CaCl_2_ × 2H_2_O, 0.5 mM MgCl_2_ × 6H_2_O, 25 mM NaHCO_3_, 10 mM HEPES, 15 mM sodium lactate, 0.25 mM sodium pyruvate, 5 mM glucose, 3 mg/mL BSA, and 50 µg/mL gentamycin for 5 min at 38.8 °C in a humidified atmosphere with 5% CO_2_ in 95% air. Then, the oocytes were extensively washed and transferred in CR1aa medium [39] supplemented with 2 mM 6-DMAP for 4 h in the same atmospheric conditions. After 4 h, oocytes were transferred into 0.5 mL of fresh CR1aa medium and cultured in vitro at 38.8 °C. All in vitro cultures were performed in 4-well plates (Nunc, Denmark) containing 0.5 mL of medium covered with mineral oil. At 32 h of in vitro culture development, the oocytes were cleaved into 2, 3, or 4 cells and collected.

### 2.3. Transmission Electron Microscopy (TEM)

Presumptive zygotes and cleaved embryos were recovered 24–28 h and 30–36 h after in vitro fertilization, respectively. All embryos were fixed in a solution of 4% paraformaldehyde (TAAB Laboratories Equipment Ltd., Aldermaston Ward, UK) and 1% glutaraldehyde (Electron Microscopy Science, Hatfield, PA, USA) in 0.1 M phosphate buffer (pH 7.4) for 1 h, washed 3 × 10 min in 0.1 M phosphate buffer, and post-fixed for 1 h with 2% osmium tetroxide (Electron Microscopy Science, Hatfield, PA, USA) in 0.1 M phosphate buffer. After washing in 0.1 M phosphate buffer for 10 min and in distilled water 2 × 10 min, samples were dehydrated in a graded series of ethanol solutions (50% ethanol 2 × 10 min, 70% ethanol 3 × 15 min, 90% ethanol 3 × 20 min, and 100% ethanol 3 × 20 min). Final dehydration was performed in 100% propylene oxide (PrOx, TermoFisher GmbH, Kandel, Germany) 3 × 20 min. Then, samples were incubated in a 2:1 PrOx/EPON epoxy resin (Sigma-Aldrich, Burlington, MA, USA) mixture with closed caps for 2 h, a 1:2 PrOx/EPON epoxy resin mixture with open caps for 16 h, and in 100% EPON for 24 h at room temperature. Samples were then replaced in new 100% EPON, incubated for 24 h at 37 °C, and polymerized for 48 h at 60 °C. “Pyramids” of minimal size were prepared using a stainless microtome blade (Feather Safety Razor Co., LTD., Osaka, Japan). Using a “Leica Ultracut UCT” ultramicrotome (Leica Microsystems GmbH, Wien, Austria) equipped with a diamond knife (Diatome, Nidau, Switzerland), 1200–1595 serial, ultrathin sections (thickness 70 nm) were cut for each zygote and embryo. Sections were placed on TEM nickel one-slot grids (Agar Scientific, Ltd., Stansted, UK) coated with Formvar film prepared from 0.25% solution of Formvar powder (Serlabo, Paris, France) in 100% 1,2-dichloroethane (VWR BDH Prolabo, Evreux, France). Sections were stained with 5% uranyl acetate (Merck, Darmstadt, Germany) in distilled water for 20 min and lead citrate for 5 min, and were then observed at 100 kV with a Jeol 1011 TEM (JEOL, Tokyo, Japan) connected to a Gatan RIO 9 digital camera driven by Digital Micrograph software (GMS 3, Gatan, Pleasanton, CA, USA). After converting to TIFF format, final versions of images were prepared with Adobe Photoshop CS3 and PowerPoint 2016 software.

### 2.4. Gene Expression

Transcriptome data on bovine preimplantation embryonic development were retrieved from the NCBI Gene Expression Omnibus database (GEO, https://www.ncbi.nlm.nih.gov/geo/ (accessed on 4 May 2020); GEO dataset accession GDS3960). Mean expression values of genes *PLK4*, *SASS6*, and *CEP192* were calculated from the values reported from two independent microarray hybridizations of mRNA from bovine oocytes, zygotes, 2-cell (2C), 4-cell (4C), 8-cell (8C), 16-cell (16C), morula, and blastocysts. Gene expression values at different stages of embryo development were presented as a histogram.

### 2.5. Measuring the Length of Centrioles

Centriolar length was measured directly using a scale bar in photographs obtained by TEM on the sections, in which the orientation of the long axis of the centrioles was near-parallel to the plane of the section. When the sections were perpendicular to the long axis of the centriole, centriolar length was calculated using the number of sections in which centrioles were observed (each section was 70 nm thick). In cases of oblique sections of the centrioles, the projection length of the MT wall of the centrioles was measured first. Next, the length of this MT segment was calculated as the hypotenuse of a triangle, in which one of the legs was set equal to the section thickness (70 nm), and the second leg was set equal to the length of the projection visible in the photographs (Figure 1).

Since the large and small triangles are similar (their corresponding sides are parallel to one other), the ratio of the lengths of their sides is proportional. Therefore, the height of an incomplete section (h) can be calculated from the ratio of the projection lengths on both full and incomplete sections (Figure 1). Next, the length of triplet microtubules in the outermost section (l) was calculated based on the incomplete height and projection (p). The total length of the triplet microtubules of the centriole was the sum of the lengths on all complete sections and one or two incomplete extreme sections (Figure 1). The angle of inclination of the long axis of the centriole to the section plane was determined from the value of tangent, which was equal to the ratio of the length of the opposite leg (the thickness of the section is 70 nm) to the length of the adjacent leg (the centriolar MT projection to the section).

## 3. Results

### 3.1. The Pre-Convergence, Two Pronuclei Zygote Has the Sperm-Typical PC and -Atypical DC

Five cattle embryos were examined on ultrathin serial sections 24 h after in vitro fertilization. In one embryo, the sperm flagellum was absent, and no centrioles were detected; effectively, this oocyte was not fertilized. In each of the four other embryos, one sperm flagellum was detected, indicating that they were normally fertilized. Of the four fertilized embryos, two had male and female nuclei separated by a large distance from each other, whereas two embryos had closely positioned pronuclei. Based on these observations, we can trace the details of the microstructural modulations in the sperm neck and reconstruct the dynamics of this process.

The major events that occur in the zygote before the first division are decompaction of sperm nucleus chromatin, detachment of the nucleus from the sperm neck region, formation of astral MTs, migration of the two pronuclei to each other, and beginning of mitotic chromatin compaction within the pronuclei. Below, we describe our vision of the ultrastructural changes that take place in the sperm neck region along with these events.

While sperm nuclear chromatin is decondensed and becomes indistinguishable from the chromatin of the female nucleus, the two pronuclei are located at a distance approximately equal to their diameter (Figure 2a,e).

The sperm flagellum, which penetrated the oocyte with nucleus, is located in the zygote cytoplasm, and its neck is separated from the male nucleus located about 5 µm away (Figure 2b). The ultrastructure of the apical (neck) region of the flagellum remains almost identical to that of the sperm neck region before fertilization: the proximal centriole is surrounded by electron-dense segmented columns, which are contiguous with dense peripheral fibers (Figure 2d and Appendix A). The apical region of the flagellum (atypical distal centriole) has the shape of an expanding funnel inside of which are two central flagellar MTs and nine peripheral MT doublets (Figure 2d,e). In the full series of sections through the apical region of the flagellum (Appendix A), astral MTs are also visible, indicating centrosome function as a center of MT organization here. We did not find any signs of degradation or duplication of the proximal centriole at this stage. The length of the proximal centriole was 405 nm.

The flagellar axoneme in part of zygotes decayed into two parts: six bundles of doublet microtubules plus two central microtubules (i.e., the formula of 6 × 2 + 2), and three bundles of doublet microtubules with no central microtubules (i.e., 3 × 2 + 0), the apical ends of which diverged over a considerable distance. The apical ends of dense fibers also diverged over a large distance (Appendix A).

Later, the male and female pronuclei approached each other (Figure 3a), accompanied by a dramatic change in the structure of the apical region of the flagellum. Segmented columns disappeared, dense fibers diverged to the sides, and their ends were observed at a considerable distance (more than 1 μm) from the proximal centriole (Figure 3b). In another zygote studied at this stage, the section through the proximal centriole passed along its long axis, facilitating measurement of its length at 440 nm.

Near the proximal centriole, we observed the formation of a structure appearing to be a dense strand approximately 400 nm long and 50 nm in diameter, surrounded by granular material arranged in rows (SB in Figure 3c–e and Appendix A). MTs of the apical region of the flagellum are disorganized and no longer form the axoneme; clusters of granular material are also found near them (Figure 3c,d and Appendix A).

At this stage, the described complex structure is the center of organization of numerous astral MTs, indicating this complex function as the zygote centrosome. No structures resembling the procentriole were found near the proximal centriole in the focus of the MT star and its environs. In each zygote studied, only one typical centriole was found: the proximal centriole of the spermatozoon.

### 3.2. The Prophase Zygote Has a Typical PC and an Atypical DC

Three embryos, 28 h post-fertilization, were examined on ultrathin serial sections. In one embryo, the sperm flagellum was absent, and centrioles were not detected. We concluded that this oocyte was not fertilized. Two other embryos had nuclei of the male and female pronuclei with varying degrees of condensed chromosomes, which we defined as varying stages of prophase of the first embryonic cleavage.

During the next stage of embryonic development, after pronuclei convergence, male and female pronuclear chromosomes began condensing, and the zygote entered prophase of the first cleavage (Figure 4 and Appendix A). In all zygotes studied, 28 h after fertilization, male and female nuclei did not differ in the morphology of their chromatin structure.

As in the earlier stages of development, only one typical centriole was found in the zygote: the spermatozoan proximal centriole of length 285 nm. In contrast to earlier stages, the pericentriolar material around this centriole has a much less dense structure, and the centriole itself has a shorter length. The apical end of the flagellum (the atypical DC) was located near the proximal centriole and contained an accumulation of granular material. As the zygote advanced from early prophase to prometaphase, the amount of electron-dense material near the proximal centriole progressively decreased.

Around the apical end of the flagellum and proximal centriole, a large number of MTs were observed; since they were obviously MTs of the aster, this indicates that they were acting as atypical centrioles, forming a centrosome.

In summary, centriole duplication with typical procentrioles seems to not occur up to the prophase stage of the first embryonic division.

### 3.3. The Prometaphase Zygote Has One Centriole and a Striated Body in the Flagellum-Associated Pole and Polar Corpuscles in the Opposite Pole

Two zygotes at 30 h after fertilization at prometaphase of the first cleavage were examined on serial ultrathin sections (Figure 5, Figure 6, Figure 7). In contrast to mitotic prophase, in prometaphase, the nuclear envelopes of both nuclei disappeared, the chromosomes were compacted, kinetochores (Figure 5d) were visible on their surface, and two poles of the forming mitotic spindle were observed, though their structure differed.

In one zygote, the centrosome contained proximal centriole (Figure 6) was detected at a distance of about 4 µm from the apical end of the sperm flagellum (see Figure 5e and Figure 6f). In the second studied zygote, the apical end of the flagellum was located near the proximal centriole, as was observed previously during mitotic prophase (Figure 4c).

The pole located nearer the apical end of the flagellum (flagellum-associated pole) contained one centriole (length 488 nm) and a striated body (Figure 6). Probably, the observed centriole was the spermatozoan proximal centriole. The striated body had an elongated shape (Figure 6f–h) and contained MT bindles (Figure 6f). A very large number of MTs were in this pole area in all the sections, suggesting it acted as a centrosome (Figure 6). The second pole of the mitotic spindle (the pole opposite the flagellum-associated pole) was also associated with numerous MTs but did not contain centrioles (Figure 7).

This consisted of two rounded, interconnected structures, about 300 nm in diameter, that were ultrastructurally similar to striated bodies. Here, we suggest use of the original term to refer to these structures, “polar corpuscles”, which was proposed in 1876 by Van Beneden, one of the discoverers of the centrosome [40]. The “polar corpuscles” were surrounded by a large number of MTs in all the sections, suggesting it acted as the second centrosome.

### 3.4. The Anaphase–Telophase Zygote Has Two Polar Corpuscles in Each Pole, One of Which Is Associated with the Apical Region of the Flagellar Axoneme

In the embryo, at late anaphase–early telophase of the first cleavage, the chromosomes have already diverged to two poles (Figure 8a,b), and the beginning of nuclear envelope formation is already visible around the chromosomes (Figure 8d).

At the same time, the spindle MTs were still visible near the poles, and the half-spindles had a pronounced conical shape (Figure 8c,d). Thus, the spindle was not barrel-shaped, as in the mouse zygote’s centriole-less poles. MTs were connected by chromosomal kinetochores and converged at the spindle poles (Figure 8c,d). The spindle zone had a less electron-dense structure than the surrounding cytoplasm near the spindle (Figure 8c).

The morphology of the pole near the flagellar apical region (Appendix A) and the opposite spindle pole (Appendix A) were studied in detail on 20 and 10 serial ultrathin sections, respectively.

At each pole, there were two “polar corpuscles” (Figure 8e,f, Appendix A) with circular cross-sections and elongated shapes (diameter about 275 nm, length 490–630 nm), which were found on seven to nine consecutive serial sections. They had an internal granular structure, similar to the striated body. However, dense central rods (like those observed at the prophase stage of two pronuclei or peripheral MT fibers in the flagellum-associated pole) were not observed; therefore, they did not contain centrioles. MT spindles ended near polar corpuscles or directly on their surface (Figure 8e,f, Appendix A).

Additionally, there were numerous interzonal MTs between two groups of chromosomes. Cytotomy (the formation of a constriction dividing the zygote into two blastomeres) had not yet begun in this embryo. One sperm flagellum was found in the cytoplasm of the zygote, which indicates that the oocyte was fertilized by one spermatozoon.

On the periphery of the embryo, the flagellum contained only an axoneme without dense fibers and mitochondria. In the middle region, this axoneme was surrounded by sperm mitochondria. At the apical end, the axoneme was again not surrounded by mitochondria. Peripheral MT axoneme doublets diverged (7 doublets remained in the axoneme, 2 diverged to the sides). Several (2–3) dense fibers were also visible at some distance from the axoneme near its apical end.

The apical region of the flagellar axoneme (the atypical DC location) was near the polar corpuscles of pole P1 of the mitotic spindle and came into contact with the lateral surface of one of the polar corpuscles (Appendix A). The sperm tail anterior end (Appendix A) lay about 500 nm from the polar corpuscles. MTs at the end were no longer visible; they were replaced by an accumulation of granular material in a structure similar to that of polar corpuscles.

It should also be noted that near the flagellum-associated pole, there was a spindle-shaped structure with a transverse stripe. In the pole opposite the spindle, which was not connected to the flagellum axoneme, there were also two polar corpuscles of a similar shape and ultrastructure, though somewhat smaller (Appendix A). A centriole and striated body were not found near this pole. MT spindles terminated near the polar corpuscles or directly on their surface, as shown for centrioles in the spindle of dividing somatic cells. These observations suggest the presence of two centrosomes, each with two polar corpuscles, and one of them associated with apical region of the flagellar axoneme.

### 3.5. The Two-Cell Embryo Has Two Centrioles in Each Blastomere

The complete structure of four “centrosomes” from two embryos at the two-cell stage (30 h after fertilization) were studied on serial sections (Figure 9, Figure 10, Figure 11 and Figure 12). Here, we have used the word “centrosome” loosely, since, morphologically, these structures were only partially similar to classical centrosomes of somatic cells.

In the first embryo (two-cell embryo I), we found one typical centriole with a procentriole in each blastomere. These centrioles were located near the blastomere nucleus (C1 and C2, Figure 9a), and had the same “standard” diameter of about 200 nm; however, their lengths were different.

In each pair, the mother centriole was very long—801 nm for MC1 (Figure 9c–k) and 1160 nm for MC2 (Figure 10c–k)—and was connected to the short procentriole—189 nm for pC1 (Figure 9d–g) and 234 nm for pC2 (Figure 10i–k**)**. The centrioles were located close to each other, and their orientation was close to mutually perpendicular, as far as this can be concluded based on oblique sections. The structure of these “diplosomes” was similar to that of the mother centriole + procentriole pair in somatic cells at S-phase. However, each blastomere of two-cell embryos contains only one centriole–procentriole pair, in contrast to somatic cells, which contain two such pairs at this stage of the cell cycle.

The sperm flagellum was not associated with centrosomes in either blastomere of this embryo. The flagellum lies in a narrow space between the cell membrane of blastomere 2 and the zona pellucida (Appendix A). The ultrastructure of the sperm flagellum did not undergo significant changes: MTs of the axoneme and circumferential (transverse) ribs were clearly visible in the main area, while in the middle area (Appendix A), the well-preserved axoneme was also surrounded by dense fibers and mitochondria (Appendix A).

In the second studied two-cell embryo (two-cell embryo II) 30 h after fertilization (Figure 11 and Figure 12), the structure of the centrosome was significantly different from that described above for the first embryo, although both were at the same stage of development. In the first blastomere Bl1, near a short typical procentriole of length 240 nm (Figure 11f–i), dense bundles of electron-dense material were visible, inside which MTs could be identified. However, these MT bundles did not form a typical centriole (Figure 11d–j).

One of these electron-dense bundles was cut transversely; it clearly was an MT triplet, similar to those observed on the transverse section of centrioles (Figure 11d–f, inserts). It can be assumed that the other electron-dense MT bundles were individual MT triplets too.

A short procentriole, of length 245 nm and structures identified by us as single MT triplets, were also found in the second blastomere of this embryo (Figure 12d,e). Additionally, the striated body was found in this blastomere (Figure 12), similar to that described previously for one of the spindle poles during the first mitotic prometaphse (Figure 6f–h). In two-cell embryo II, the sperm flagellum lay outside of the zona pellucida, which probably indicates that it was broken during fertilization quite close to the spermatozoan neck region (Appendix A).

### 3.6. Blastomeres of the Second Cleavage Have Atypical Centrioles

One of the embryos, in which the second cleavage division began, contained three blastomeres at 30 h after fertilization. An ultrastructural study showed that one of the blastomeres (in the cytoplasm of which a sperm flagellum axoneme was found) was at prophase of the second cleavage (Figure 13, Figure 14, and Appendix A), while the other two were at the cytotomy stage; their chromosomes had already formed interphase nuclei, but the cytoplasm was not yet completely separated (Figure 15, Figure 16, Appendix A). Thus, an ultrastructural study of this embryo could produce a picture of the structure of the spindle poles in prophase and telophase cells during the second cleavage.

In prophase of the second embryonic division, we found two centrosomes whose structures were significantly different (Figure 13 and Figure 14). At the left pole, there was a single centriole, the triplets of which were connected with an electron-dense material. (Figure 13e–h). There were many MTs near this centriole, some of which departed from the surface of the centriole, and others which ended in the material of the centriolar mitotic halo, similar to behavior observed in somatic prophase cells.

The second centrosome (right pole) contained several irregularly oriented short MT bundles, which we identified as individual separated triplets. Numerous MTs ending in the mitotic halo were also present near this centrosome (Figure 14). Appendix A shows sections before and after the series of sections shown in Figure 14.

In the telophase-cytotomy cell, which we initially identified as two separate blastomeres at light microscopy level, one centrosome was found in each of the blastomeres. The structures of these centrosomes had similar compositions.

In all three blastomeres, centrosomes contained several irregularly directed MT bundles. One of these bundles was cut perpendicularly, which allowed us to identify it as an isolated individual triplet (Figure 16c–g, inserts), similar to the one that we previously showed in a two-cell embryo (Figure 11d–g, inserts). MTs were detected near both centrosomes. Appendix A show sections of the previous and subsequent series shown in Figure 15 and Figure 16.

### 3.7. Typical Centrosomes Are Formed in Blastomeres after the Third Cleavage

Two embryos following the third and fourth cleavages were examined. One contained 7 blastomeres, and the other 14. The centrosomes in these embryos contained either two long centrioles close to each other without procentrioles (Figure 17) or two long centrioles at considerable distance from each other, with procentrioles of different lengths associated with them (Table 1 and Table 2). Atypical centrioles were found in two blastomeres in each embryo (see below).

Closely oriented long centrioles formed the diplosome. The mother centriole could be distinguished due to mutual arrangement of centriolar cylinders. Both ends of mother centrioles were “free,” i.e., not covered by the second centriole (Figure 17c–k), whereas one of the ends of the daughter centriole was directed towards the surface of the mother centriole (Figure 17f–h). According to our measurements, the mother centriole in the blastomeres was always longer than the daughter centriole, although the length of both centrioles varied widely between different blastomeres (Table 1 and Table 2). In the daughter centriole oriented parallel to the section plane, the proximal part of its lumen was filled with electron-dense material approximately 200 nm from the proximal end (Figure 17g). Obviously, this was a “cartwheel structure”, which is the basis of 9-ray symmetry during the formation of the procentriole, is preserved in the daughter centriole in vertebrates during the next cell cycle, and disappears only in the mother centriole. Since in all cases studied (Table 2, N = 6) the mother centriole was longer than the daughter centriole, we assumed for centrioles at a great distance from each other that the longer centriole in the pair was the mother centriole. Comparison of the lengths of procentrioles showed that the procentriole of the mother (longer) centriole was always longer than the procentriole of the daughter (shorter) centriole (Table 1 and Table 2).

In the 14-cell embryo, in contrast to the 7-cell embryo, blastomeres were found in which there were no procentrioles on the mother centrioles (Table 2). This centrosome morphology is characteristic of cells in the G1 phase of the cell cycle, so we can assume that this phase of the cell cycle appears after the third cleavage.

In both 7-cell and 14-cell embryos, two blastomeres contained one atypical centriole associated to a typical procentriole. In each embryo, one atypical centriole was associated to the axoneme of the spermatozoa flagellum, which remained in the blastomere cytoplasm. The second atypical centriole in each embryo was detected in the blastomeres without flagellum remnants.

The structure of the atypical centrioles was similar to that described in early embryos. Irregularly arranged beams (triplets) of MTs were associated with electron-dense material (Figure 18g,i). The atypical centriole was located close to the apical end of the sperm flagellum (Figure 18 and Appendix A). The ends of the axoneme peripheral MT doublets diverged from each other. At the same time, the second centrosome in this blastomere had a structure typical of somatic cells (Figure 19). On the cross-section of the centriole, the “cartwheel structure” was clearly visible (Figure 19i, insert).

The typical centriole in this blastomere was 644 nm long and 200 nm in diameter (Figure 19d–g); its proximal end was not filled with an electron-dense material, as expected, from a mature centriole (aka mother centriole) (Figure 19f).

All typical centrioles in the 7- and 14-cell embryos lacked distal and subdistal appendages; striated rootlets and centriolar satellites were also not found near them (Figure 19d–g). Many MTs were observed in the centrosomal region, and the pericentriolar material did not have significant electron density.

Because a typical procentriole was formed on an atypical centriole in embryos at earlier stages, we can conclude that the oldest centriole from the blastomere in the 14-cell embryo was atypical, while a typical centriole formed on this atypical centriole one cell cycle prior.

This blastomere was unique among the 14 cells of this embryo because it contained the sperm axoneme. The structure of the centrosome allows us to conclude that an atypical centriole with an irregular organization of triplets is capable of producing centrioles with normal structure, even after the first three or four cell cycles of embryonic development.

It is interesting to note that the developing procentriole was more often located on the mother centriole, on the side of the blastomere nucleus. From 13 procentrioles in the 7-cell embryo, where we could estimate the location of procentrioles relative to the nucleus, 8 procentrioles were located between mother centriole and the nucleus, 4 were located beside the mother centriole, and, in only 1 case, the mother centriole was found between the nucleus and procentriole. Of 13 procentrioles in the 14-cell embryo, 9 were located between mother centriole and the nucleus, 5 were located beside the mother centriole, and in only 2 cases, the mother centriole was found between the nucleus and procentriole. Thus, for these 2 embryos, the ratio of procentrioles located on the nucleus side to those located on the opposite side was 17 to 3, with 9 centrioles located on the mother centriole side.

### 3.8. Centriole Appearance in Parthenogenetic Embryos

Two four-cell parthenogenetic embryos were examined in complete series containing 1561 ultrathin sections, with only 6 sections missing (sections 226, 566, 1057, 1058, 1462, and 1463). The absence of the sperm flagellum in the embryonic blastomeres confirmed their parthenogenetic development. A centriole spans at least three to four sections, even if its orientation is exactly parallel to the cut plane, so it would be impossible to overlook the centriole using serial sections through the blastomere in our experiment.

In all blastomeres of one four-cell embryo, neither centrioles nor their precursors were found. Thus, in this embryo, the first two divisions took place without the participation of centrioles. One blastomere of this embryo was binuclear, which may indicate a violation of the normal distribution of chromosomes among blastomeres.

In the second embryo, no centrioles were found in the three blastomeres. In one cell, a single, long centriole was observed (Figure 20). The centriole was detected through 12 consecutive oblique serial sections, and the triplets of its wall had unequal lengths, ranging from 1014 to 1486 nm (Figure 20j–o). Numerous cytoplasmic microtubules were located around this centriole. A large, striated rootlet was found in one blastomere (Figure 21).

### 3.9. Centrosome-Associated Protein Gene Expression in Bovine Embryos

To analyze expression patterns of the genes coding for centriole-associated proteins PLK4, SASS6, and CEP192 in early bovine embryos, we used GEO datasets (accession GDS3960) on preimplantation embryo development in bovine [41].

As demonstrated (Figure 22), *PLK4* mRNAs are present at all stages from the oocyte on, with transcript levels decreasing in the 4-cell embryo and transcription being reactivated in the 8–16-cell embryo.

*SASS6* expression increases starting in the eight-cell embryo. *CEP192* showed lower abundance in the 4-cell embryo, with temporal reactivation in the 16-cell embryo.

## 4. Discussion

During embryo development in cattle, the transition from the zygote—which has only one proximal centriole, derived from the spermatozoon—to the centrosomal structure observed in somatic cells is a sequential process, characterized by gradual regulation of centrosomal and centriolar proteins in the blastomeres. It is likely that decreases in the volume of the blastomeres at each cleavage progressively contribute to finer regulation of this process.

We will begin our discussion of the results with an analysis of the appearance of centrioles during parthenogenetic development. Such an analysis will immediately set apart the events and effects associated with the penetration of the sperm into the oocyte and the introduction into the zygote of its neck region, which includes the proximal centriole, the segmented column material, and other components associated with the end of the flagellum adjacent to the nucleus.

### 4.1. Centrioles in Parthenogenetic Embryos

Only a single centriole was found for all eight studied blastomeres from two parthenogenetic embryos; still, this one centriole provided a great deal of information. Firstly, it is now clear when the formation of de novo centrioles begins in bovine parthenogenetic embryos; secondly, it is clear that this formation occurs asynchronously in different blastomeres (as in mice during normal development [15]); thirdly, one centriole appears first (again, the same as in mice [15]); fourthly, this centriole is comprised of triplets of unequal length, which is also similar to what we observe after fertilization; fifthly, we see that de novo centrioles begin to form only shortly after fertilization or artificial activation of development; sixthly, no precursors to centriolar formation were found in the blastomeres, which means that precursors originate from the material of the spermatozoon.

### 4.2. The Disappearance of the Proximal Sperm Centriole in the Zygote and Appearance of New Typical or Atypical Centrioles in Two- to Four-Cell Embryos

In the present work, ultrastructural studies helped to characterize the morphology of developing centrosomes in zygotes and cleaved embryos. The present study was carried out on bovine zygotes and early embryos following in vitro fertilization, which made it possible to compare the electron microscopy data presented here to immunofluorescence data obtained by the Avidor-Reiss laboratory [32].

A detailed study of bovine zygotes by immunofluorescence using specific antibodies showed that, despite the loss of typical centriole morphology, the distal centriole was found at the apical portion of the flagellar axoneme, and it retained centriolar proteins [32]. During the first zygotic cleavage, a pair of foci stained with antibodies to centriolar proteins was observed in each of the poles. At the same time, one of the poles adjoined a more elongated structure, identified by the authors as the sperm flagellum axoneme. The authors interpreted an observed doubling of the foci stained with antibodies to centriolar proteins as centriole duplication. They hypothesized that the proximal centriole of the sperm serves as a platform for the duplication of one centriole, whereas the “atypical” centriole, which originates from the head of the flagellum axoneme, serves as a platform for the formation of the second procentriole [32]. Therefore, they proposed that only one cycle of centriole duplication occurs in the bovine zygote.

Other published data is consistent with the observed presence of some typical centrioles post-fertilization [22,33]. Thus, in sheep, in three out of thirteen poles studied, typical centrioles were not found in the first spindle, whereas three centrioles were found in two spindles, and at least one centriole per spindle was found in the remaining eight studied poles as reported [22]. In human zygotes at the pronuclei stage (only for bipronuclear embryos), typical centrioles were found in only one of three studied embryos, and at the syngamy stage, one cell of two studied contained typical centrioles (Table 1 in [33]). These data clearly indicate that typical centrioles can present in, but are not required components of, the centrosomes and spindle for the first embryonic division.

The disappearance of the proximal sperm centriole in the zygote is not something exceptional. After all, such a mechanism of centriole elimination exists and is realized during oogenesis; after fertilization, it can probably be partially preserved. In addition, one should not forget that the distal centriole of the spermatozoon is also resorbed during spermiogenesis.

Moreover, in the mouse early embryo, only one typical centriole appeared de novo in the first blastomeres [15], whereas classical centrosomes with two centrioles appeared gradually in the blastocyst blastomeres. However, unlike the barrel-shaped spindles of the first embryonic cleavage in mice, “classical”, spindle-shaped spindles in bovine, as observed in somatic cells, indicated the possible presence of centrioles in embryos.

### 4.3. What Was Found at Division Poles of Bovine Blastomeres?

Two structures marked by centriolar protein CEP152 were detected by immunofluorescence at each of the poles during the first division of the bovine zygote [32,42]. One of these four structures was larger than the others and was associated with the axoneme of the sperm flagellum. However, CEP152 and SASS6 were not detected in spermatozoa [32]. Based on ultrastructural analysis of sperm, the authors suggested that such “atypical centrioles” could derive from the sperm distal centriole, which was remodeled during spermiogenesis. This remodeling is necessary to endow the sperm neck with new functionality as a dynamic basal complex that mechanically links the sperm tail to the head [43].

In humans and cattle, two functional centrosomes are presumed to be derived from either the proximal centriole or the distal centriole [44,45,46]. In the present work, using the full series of ultrathin sections through individual zygotes with distant pronuclei, we have studied in detail the dynamics of changes to the apical part of the sperm flagellum. In the zygote, just after the loss of the connection between the sperm nucleus and flagellum, the structure of the flagellar end was similar to that of the free spermatozoan neck. Later, dense fibers and segmented columns were disaggregated and transformed into granular material associated with the proximal centriole.

We did not observe the formation of new typical centrioles in any of the eight analyzed zygotes. The proximal centriole looked deformed in prometaphase, and we did not find typical centrioles at telophase of the first mitotic division (Figure 23). During the transition from zygote with distant pronuclei to the stage of proximal pronuclei, the striated column material was transformed into a structure that we referred to as the “striated body”. This structure was a center of MT organization in the zygote and was probably the source of appearance of polar corpuscles, which were found at the poles during the first mitotic division in prometaphase and telophase cells (Figure 7, Figure 8, Figure 23, Appendix A).

Since we observed some typical centrioles in the embryos at the two-blastomere stage, questions arose as to how they were formed and what was their origin.

The length of both centrioles with canonical-like structure that we found in one of the two-cell embryos was twice that of the spermatozoan proximal centrioles observed in zygotes. Additionally, these long centrioles had triplets of unequal length, indicating that they were likely in the process of formation. These observations allowed us to conclude that these centrioles were not proximal centrioles from the spermatozoon but originated de novo.

In another two-cell embryo and in the embryo at the telophase/cytokinesis stage after the second division, we found irregularly located MT triplets, which can be qualified as atypical centrioles (Figure 23). A similar structure, bearing irregular number and orientation of triplets, was observed during centriole formation in cells mutant for the SASS6 protein [47].

According to gene expression patterns of *SASS6*, *CEP192*, and *PLK4* during early embryo development, both CEP192 and PLK4 demonstrated degradation of maternal mRNA pool at four-cell embryo, and reactivation of transcription at eight-cell embryo, when major genome activation starts in bovine embryo. Therefore, in four-cell embryo, there is likely a shortage of *PLK4* and *CEP192* transcripts that may lead to lack of corresponding proteins and, consequently, “unbalanced morphogenesis” of centrioles observed at stage 2 and 4 blastomeres.

Our data showed that typical centriolar formation was not synchronized during the stages of embryonic development. In fact, only one centriole was found in one of the poles of the prophase cell in the embryo after the second cleavage (Figure 13 and Figure 23), whereas no fully formed centriole was observed in the second pole of this blastomere; rather, only irregular MT triplets, i.e., an atypical centriole, were observed (Figure 14 and Figure 23).

Unlike previous studies, in this study we were able to analyze the entire volume of the centrosomes and surrounding cytoplasm on serial sections, thus eliminating the possibility of “unobserved centrioles” and misinterpreted results from a single section.

It is important to note that normal procentrioles were always formed on atypical centrioles. Thus, during embryonic development, atypical centrioles were, functionally, full-fledged templates for new procentrioles in each subsequent cell cycle, but did not transform into canonical centrioles themselves.

Consequently, there is no fundamental difference in the mechanism of centriole formation in mice and cattle. However, typical centrioles are formed at different stages of embryonic development in cattle and mice: two and greater than eight blastomeres, respectively. Thus, in the bovine zygote, the interval between the possible disappearance of the typical structure of the proximal sperm centriole and the appearance of new typical centrioles at the two-blastomere stage is very short. This may explain the observed presence or absence of centrioles at these stages reported in earlier studies, as well as the intermittent presence of only one centriole at the spindle pole in human and sheep embryos [22,32,33].

To build a cellular structure as complex as a centriole, coordinated and precisely regulated activation of a whole gene orchestra is required. In somatic cells after mitosis, a centrosome contains two centrioles, and in the second half of G_1_ phase, new procentrioles appear on each centriole [48,49,50] and achieve the same size as mother centrioles at the beginning of mitosis. In early embryogenesis, the process of such a transition is delayed by several cell cycles, probably due to the shorter duration of the cell cycle in embryos than in somatic cells.

Various authors have reported the appearance of the first typical centrioles in early embryos of mice, rats, and rabbits at various stages of development: from 8–16 blastomeres to the blastocyst [7,12,13,14,15,24,51,52]. In our opinion, this is not due to experimentation accuracy but is rather reflective of real variability in the timing of this event in mammals. Indeed, at these early stages of development (from zygote to blastocyst), a cilium or flagellum is not required, and, thus, embryonic cells may not have a special need for typical centrioles.

Similarly, in cattle, sheep, and humans, the blastomeres from the first embryonic divisions do not require “classical” somatic-type centrosomes. The formation of atypical centrioles in these cases may be a preparation for subsequent phases of embryonic development, during which the centrioles would be necessary. Centrioles with typical structure are necessary to form cilia during the formation of embryonic right–left asymmetry [53,54]. Before this time, the embryo does not need to expend extra energy and “building materials” for the construction of typical centrioles. Similarly, cells of the wasp *Anisopteromalus* have no cilia or flagella at the early stages of larvae development, and their centrioles lack MT triplets; however, the centrioles of adult wasps acquire a “normal” structure comprised of nine MT triplets [55]. A similar situation of centriolar structural diversity was also reported in *Drosophila*, in which the centrioles in cilia-free somatic cells consisted of MT doublets, whereas ciliated germ cells and ciliated sensory cells had centrioles consisting of MT triplets [56].

During embryonic development, centriolar proteins have specific gene expression patterns. The presence of scattered, irregularly oriented triplets of MTs—instead of centrioles—reported here may mean that expression of centriolar protein genes differs between the blastomeres during the first stages of embryogenesis. This agrees with the significantly different centrosomal structures observed between different embryos at the same stage (Figure 23 and Figure 24) and between different blastomeres of the same embryo. Additionally, a deficiency in molecular regulation might be associated with the huge volume of blastomeres as compared to the volume of somatic cells, which may affect physical interactions between intracellular structures. Decrease in cell size during cleavage may facilitate the accuracy of molecular regulation.

Formation of defective centrioles with an irregular number and orientation of triplets in SASS6 protein knockout cells was previously described [47]. The structure of “error-prone” centrioles in these cells (Figure 4 in [47]) was similar to that of the atypical centrioles in two- and four-cell embryos reported here (Figure 11, Figure 12, Figure 14, and Figure 16). This may mean that formation of the first centrioles during early embryonic development in cattle can occur despite SASS6 protein deficiency. Structurally similar individual triplets that do not form centrioles were also found upon loss of SASS6 in mouse embryonic stem cells in vitro [57].

The low level of SASS6 expression in cattle embryos persists until the eight-blastomere stage, when the embryonic genome is activated [58]. This is consistent with our hypothesis that the appearance of individual MT triplets, which did not form centriolar cylinders at the two- or four-blastomere stage, may be associated with low expression levels of SASS6 protein. SASS6 is a key regulator in the formation of normal centrioles with nine-fold symmetry of triplet microtubules [59,60]. At the same time, transcripts of two other proteins involved in the formation of centrioles are more abundant in the zygote and in two-cell embryos. As a result, centrioles may be formed with defective triplet organization, similar to that described for SASS6 knockout cells [47]. One way to interpret this observation is that centrioles are formed de novo, with some randomness in different blastomeres, and in the process of their formation they undergo phases of unbalanced morphogenesis.

### 4.4. Retrospective Analysis of Centriolar and Centrosomal Formation during Early Embryonic Development According to Morphological Analysis of Different Blastomeres in Bovine Embryos

For the first time, we observed blastomeres with four centriolar cylinders (two centrioles and two procentrioles) in a seven-cell embryo. At this stage, SASS6 expression increased and may be involved in the control of somatic centriolar organization. However, all studied centrioles lacked additional structures usually associated with them, including distal appendages, subdistal appendages, and striated rootlets. This means that the formation of full-fledged somatic centrosomes is not yet completed by this stage, and these structures may appear at later stages of embryonic development.

As previously shown, despite the absence of centrioles at the poles during the first cleavage, centrosomal proteins such as γ-tubulin [9] and centriolar protein CEP152 [32] are present at the spindle poles. Ultrastructural study of the poles of the first cleavage spindle prompted us to suggest that polar corpuscles may serve as concentrators of these proteins to the poles during the first cell division in cattle. The composition of the striated body found near the apical end of the flagellar axoneme proximal to one of the first division spindle poles and in the centrosome at the two-blastomere stage remains unclear. This could be a site at which centrosomal proteins concentrate in the absence of centrioles.

The blastomeres in the 7- and 14-cell embryos were at different stages of the cell cycle, as evidenced by the varying lengths of their procentrioles. In those blastomeres in which procentriole growth has not yet begun, two closely aligned centrioles were observed. Through their mutual orientation, it was possible to determine which of them was mother centriole (both ends were “free”) and which was daughter centriole (proximal end was directed towards the second centriole). The mother centriole was always longer than the daughter centriole, as shown in 6 blastomeres of a 14-cell embryo. In contrast to centrioles in which the process of duplication had not yet begun, centrioles with procentrioles were always localized at considerable distance from each other. This indicates the close relationship between the divergence of centrioles and the timing of their duplication initiation.

However, even in separated centrioles with procentrioles, one centriole was significantly longer than the second centriole in the same cell (Table 1 and Table 2). Based on the data on centrosomes with non-separated centrioles, where the longer centriole was always more mature (mother centriole), we can conclude that, in other blastomeres as well, the longer centriole can be considered the mother centriole. The maximum length of procentrioles before division can be estimated from the longest procentriole (518 nm) and the shortest daughter centriole (574 nm) observed here. Since the length of the mother centriole also significantly varied (652–1059 nm), it may be suggested that strict regulation of centriolar length in somatic cells has not yet been established in embryos at these stages of development.

One of the traditional methods for studying the function of proteins is blocking their synthesis or, conversely, stimulating their overexpression, followed by observation of changes in the functioning of various cellular components. The most illustrative example of an application of this type of analysis is our assumption that the lack of formation of the centriolar cylinder in the presence of randomly oriented triplets is caused by the deficiency of SASS6 protein [47]. In this study, due to our observation of both centriolar and centrosomal formation, we may hypothesize that expression levels of centrosomal proteins may affect centriole appearance. Data on low levels of SASS6 protein in two- to four-cell embryos in cattle corroborate this (Figure 22).

Additionally, it is logical to suppose that the non-standard length of maternal centrioles in different blastomeres of the same embryo and triplets of unequal length in the composition of one centriole may be the result of CPAP and CP110 protein ratios in blastomeres, which play a fundamental role in the regulation of triplet microtubule length in centrioles [61,62,63,64,65]. This may also be related to the fact that the growth of centrioles continues during the next cell cycle, so more mature centrioles are longer. It should also be noted that the length of centrioles in blastomeres was significantly greater than in somatic cells.

Special attention should be paid to the analysis of the inheritance of atypical centrioles. The data on centrosomal structure in all blastomeres of 7- and 14-cell embryos allowed a retrospective analysis of centriolar formation at earlier stages of development. As was previously shown in two- to four-cell embryos, centriolar construction is a complex process. At later stages of microtubule triplet formation, centrioles are not rigidly tied to earlier ones, as is the case during normal construction of centrioles in somatic cells. Similarly, there is no direct relationship between the formation of microtubule triplets and the formation of the cartwheel structure. The result of an imbalance between different centriolar proteins may be the formation of atypical centrioles, which are randomly oriented triplets of microtubules that are grouped together.

Analysis of centrosomes in 7- and 14-cell embryos showed that, in both cases, there were two atypical centrioles per embryo. On all four such centrioles, there were procentrioles of normal structure; at the same time, and in the same blastomere, there was a second centriole of canonical structure with a procentriole. As was shown in earlier embryos, only the canonical procentriole grows on an atypical centriole; therefore, both atypical centrioles in 7- and 14-cell embryos arose not one from the other, but independently of each other and, probably, simultaneously. As such, two atypical centrioles in both cases were formed at the two-blastomere stage, and the precursors of these structures were polar corpuscles observed during late anaphase of the first cell division. Furthermore, these atypical centrioles continued to be inherited unchangingly after more than two or three embryonic cell divisions and, at each stage, acted as mother centrioles during formation of the procentriole. Thus, it is obvious that, despite structural disturbances, an atypical centriole functions normally, can support the formation of one procentriole, and duplicates only once per cell cycle, just as in somatic cell centrosomes.

In mice and other rodents, the processes of centriolar and canonical centrosome formation are delayed. This leads to the question, have they passed the stage of parthenogenesis in their development?

It has been shown that mice have, at first, one centriole in the blastomere [15]. In one of the two-cell stage blastomeres, we also found only one centriole in the blastomere. Obviously, the efficiency of mitosis in embryos does not depend on the number of centrioles in the centrosome; it can occur efficiently with a non-centriolar pole of the spindle, a pole with one centriole, or a pole with two centrioles. They key factor is rather the presence of functionally active proteins. Centrioles are critical components of later embryonic development for building cilia and flagella. Within the centrosome, centrioles are rigid and stable due to acetylation and other post-translational modifications of tubulin—a protein component of microtubule triplets—and their association with microtubule-associated proteins. Therefore, the centrosome plays the role of platform for the accumulation and retention of many functionally active proteins. Additionally, the emerging centrosome in blastomeres is always located in the vicinity of the Golgi complex, which obviously suggests possible participation of this organelle in the maturation of centrosomal proteins.

It is also important to note that one of the atypical centrioles in both 7- and 14-cell embryos was associated with the axoneme of the sperm flagellum, which thus remained in the cytoplasm of the blastomere even after the fourth cleavage division, and often, although not in all cases, retained its connection with a centrosome. Thus, this centriole originated from the segmented column material, as we showed previously in two- to four-cell embryos, while the second atypical centriole formed de novo from polar corpuscles in the opposite pole (opposite the pole associated with the axoneme of the sperm flagellum) of the first division spindle in the zygote. Unlike fertilized embryos, parthenogenetic embryos never showed such a structure as a striated body. We suggest that the striated body is an intermediate structure between striated column material of the spermatozoan neck and the centrioles.

In the embryo after the fourth cleavage, there were two blastomeres that differed from other cells and from each other in their centrosomal structure and in the presence/absence of a sperm flagellum axoneme associated to the centrosome. During embryonic development, the differences in centrosomal development may lead to asymmetries of embryonic cells and, further, provide cause for cell differentiation. It is likely that developmental disorders in parthenogenetic embryos may be associated, among other things, with a lack of asymmetry in the blastomeres.

Different lengths of procentrioles in different blastomeres indicate a desynchronization of the cell cycle between the blastomeres and, thus, may indicate further differential development of the blastomeres. In one blastomere of a 14-cell embryo (Cell 14, Table 2), we found very significant differences in procentriole lengths: on its mother centriole, the procentriole had already grown to 350 nm, while on another, a procentriole had not yet appeared. These types of differences have never been observed in somatic cells. As already noted, such a desynchronization of centriolar duplication may be due to an imperfect signal delivery system in an extremely large volume of blastomeres, compared to somatic cells.

In blastomeres, the atypical centriole is the eldest in the pair, and its procentriole is always longer than the procentriole of the second centriole. Similarly, in other blastomeres, differences in length are observed between the procentrioles. In these cases, procentrioles on mother centrioles (more mature, originating at least one cell cycle earlier) were always longer than on second centrioles. In the cases where two centrioles had not yet separated, the mother centriole (determined by the two free ends) was always longer than the second daughter centriole “born” in the previous cell cycle. Thus, we observed structural differences between the centrosomes in different blastomeres of the same embryo and between two centrosomes in the same blastomere.

As shown here, in bovine blastomeres, the procentriole is predominantly located on the nuclear side of the mother centriole. Similarly, in *Drosophila* syncytial embryos, an assembly of new procentrioles always started on the side of the wall that faced the nuclear envelope [66]. These data indicate a specific signal originating from the nucleus that initiates centriolar duplication.

It is believed that the G1 and G2 phases are practically absent in the blastomeres during first cleavages and that S phase and DNA replication in the nuclei start immediately after mitosis. In the 7-cell embryo, all centrioles had procentrioles, so centriole duplication had begun no later than the start of DNA replication, similar to somatic cells. However, in 6 out of 14 blastomeres of a 14-cell embryo, procentrioles were absent on both mother centrioles. This may indicate that in blastomeres at this stage of development, G1 phase has already occurred, or that centriolar duplication begins after the start of DNA replication, in contrast to somatic cells [45].

The formation of atypical centrioles/polar corpuscles is probably just one of the possible scenarios that characterizes spindle pole formation during the first embryonic cleavage. Another possible scenario suggests the involvement of typical centrioles in this process [22,35]. Although preservation of the proximal sperm centriole in two-cell embryos was not supported by our data, this scenario of centrosome formation could not be excluded during embryonic development in other species [22,33,35]. In this case, the sperm proximal centriole should be duplicated; however, we have never observed this in bovine zygotes. Indisputable evidence of de novo formation of centrioles during early embryonic development in bovines was supported by the appearance of centrioles in parthenogenic embryos that were already at the four-cell stage. Therefore, formation of centrioles may occur in the cytoplasm of early bovine embryos regardless of the presence of the proximal sperm centriole and material from the entire spermatozoan neck region.

## 5. Conclusions

The data obtained here show that, in bovine early embryonic development, as in mice, typical centrioles can be formed in the absence of preexisting typical centrioles. Differences between the two species include the embryonic stage during which centrioles reappeared and the presence of atypical centrioles originating from the sperm. In cattle, typical centrioles start to appear very early, at the two- to four-blastomere stage, whereas in mice, typical centrioles appear much later, at the blastocyst stage.

It is possible to describe several stages of centrosomal formation in the blastomeres leading to canonical centrosomes of future somatic cells:

(1) The concentration of microtubule nucleation proteins in the poles of the first division spindle and the formation of polar corpuscles, which are non-centriolar centers of MT nucleation and organization;

(2) Formation of microtubule triplets, which at the first stage are not organized into centriolar cylinders (atypical centrioles) but are already capable of concentrating various initiation and growth factors of canonical procentrioles on their surface. The unbalanced morphogenesis of centriole may be associated with an insufficient amount of the principal centriolar proteins CEP192 and PLK4 at this stage of development;

(3) Formation of centriolar cylinders with triplets of unequal lengths, due to potential dysregulation of the ratio of centriolar proteins;

(4) Formation of canonical centriolar cylinders with triplets of the same length;

(5) Appearance of additional structures associated with centriolar cylinders, i.e., distal and subdistal appendages, and the formation of the primary cilium.

## Figures and Tables

**Figure 1 cells-12-01335-f001:**
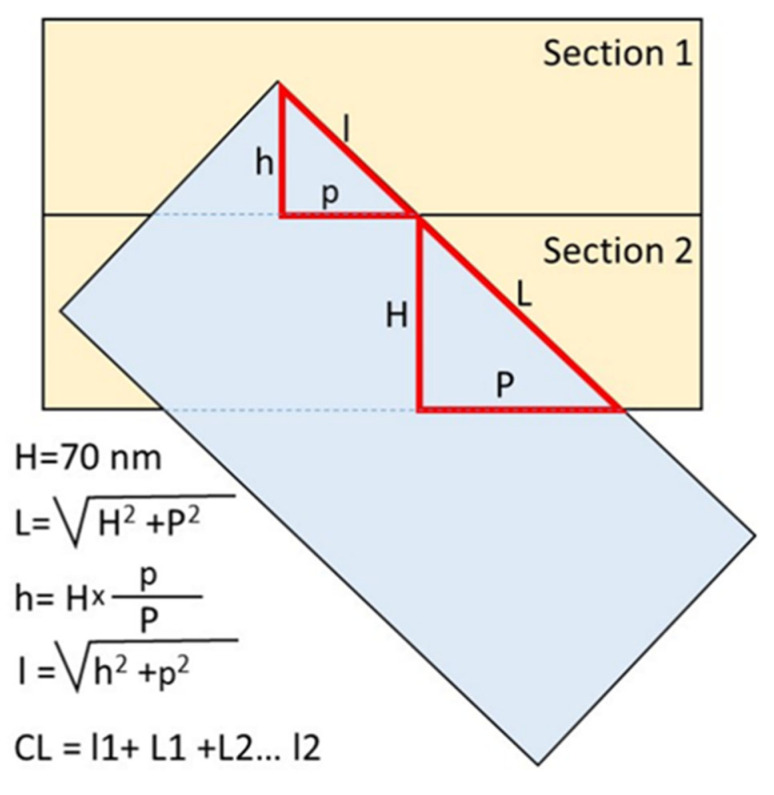
Measurement of centriolar microtubule triplet length on oblique sections. The projection lengths p and P were measured on TEM photographs; the height H (section thickness) was 70 nm. The total length of the centriolar cylinders was the sum of all complete sections and the first and last incomplete sections in which the centriole was present. CL—total centriolar length.

**Figure 2 cells-12-01335-f002:**
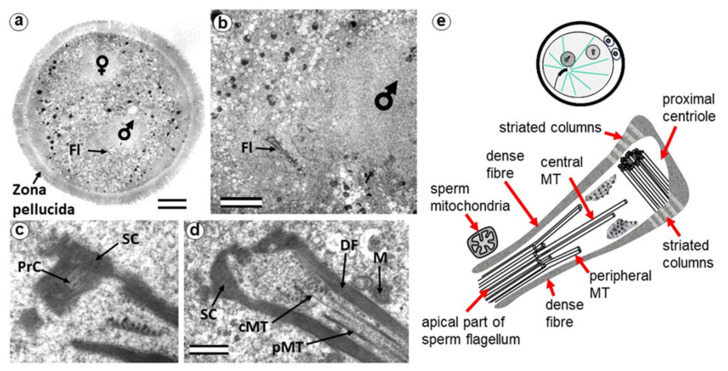
The zygote with two pre-convergence pronuclei has a typical PC and atypical DC (24 h after fertilization). (**a**) General view of the zygote at low magnification; sperm and egg nuclei are designated as male and female, respectively; (**b**) apical region of the sperm flagellum near male pronucleus; (**c**,**d**) two sections at high magnification; (**e**) schematic diagram of the structure of the centrosome at this stage. cMT—central MT of flagellum; DF—dense fibers; Fl—flagellum; M—mitochondria of the sperm; PrC—proximal centriole; pMT—MT of peripheral doublets; SC—striated column. Scale bars: (**a**)—20 µm; (**b**)—5 µm; (**c**,**d**)—0.3 µm. All 10 serial sections at high magnification are presented in Appendix A.

**Figure 3 cells-12-01335-f003:**
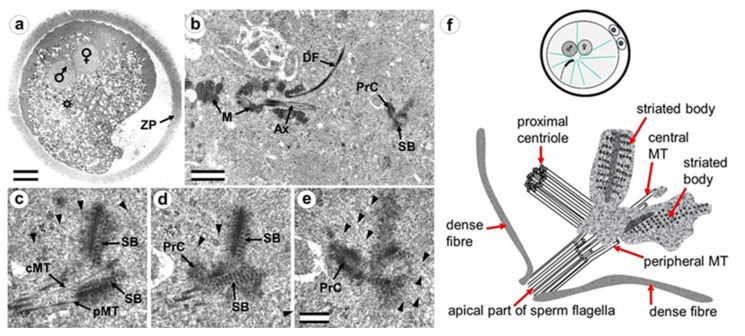
Zygote at the close pronuclei stage (24 h after fertilization) has the sperm-typical PC and -atypical DC. (**a**) General view of the zygote at low magnification; sperm and oocyte nuclei are designated as male and female, respectively, and an asterisk indicates the position of the apical end of the spermatozoan flagellum; (**b**) apical region of the sperm flagellum near male pronucleus; (**c**–**e**) three serial sections at high magnification; (**f**) schematic diagram of the structure of the centrosome at this stage. cMT—central MT of flagellum; PrC—proximal centriole; pMT—MT of peripheral doublets; SB—striated body; ZP—zona pellucida; arrowheads indicate MTs in cytoplasm. Scale bars: (**a**)—20 µm; (**b**)—1 µm; (**c**–**f**)—0.2 µm. All 9 serial sections at high magnification are presented in Appendix A.

**Figure 4 cells-12-01335-f004:**
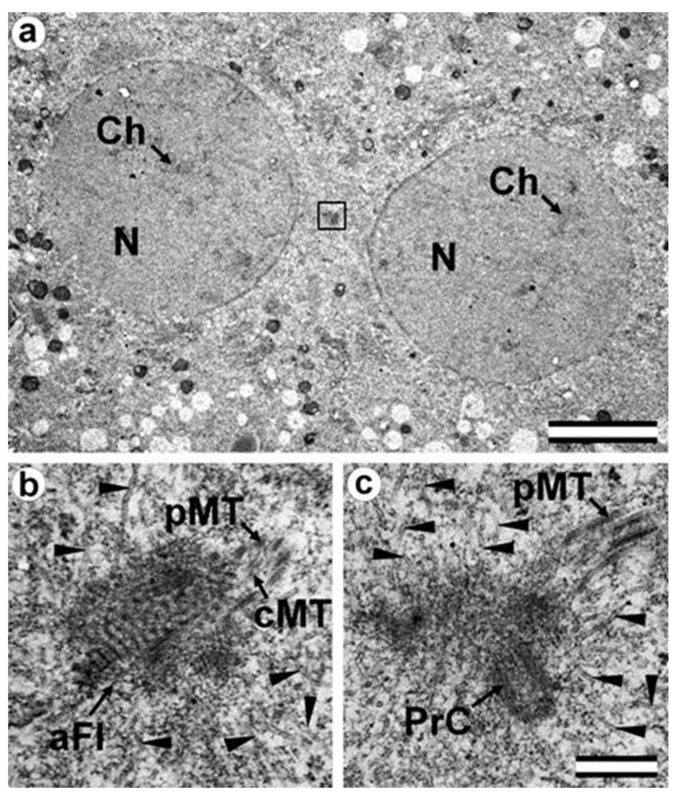
The early prophase zygote has a shorter sperm-typical PC and -atypical DC (28 h after fertilization). (**a**) General view of the apical region of the flagellar region; (**b**,**c**) two of twelve serial sections at high magnification with the apical region of the flagellum (**b**) and proximal centriole (**c**). aFl—apical region of flagellum; cMT—central MT of flagellum; Ch—chromosomes; N—nucleus; pMT—MT of peripheral doublets; PrC—proximal centriole; arrowheads indicate MTs in cytoplasm. Scale bars: (**a**)—5 µm; (**b**,**c**)—0.3 µm. All 12 serial sections at high magnification are presented in Appendix A.

**Figure 5 cells-12-01335-f005:**
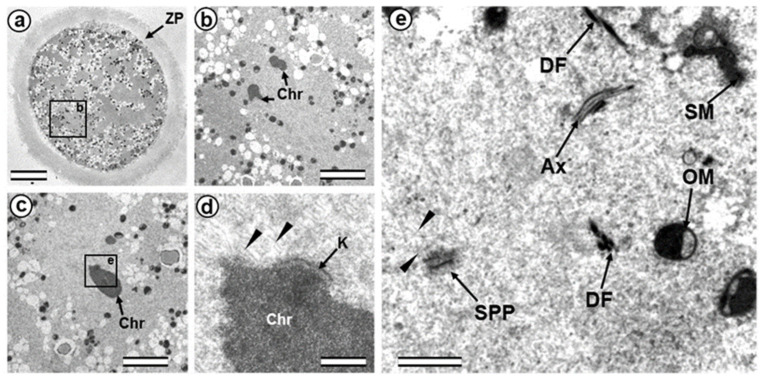
The prometaphase zygote (30 h after fertilization). (**a**) General view of the zygote at low magnification; (**b**) spindle region; (**c**) one of the chromosomes; (**d**) kinetochore region of the chromosome shown in (**c**); (**e**) apical part of axoneme and flagellum-associated pole of spindle. Ax—axoneme of sperm flagellum; Chr—chromosomes; DF—dense fibers of sperm flagellum; K—kinetochore; OM—mitochondrion of oocyte; SM—mitochondrion of sperm; SPP—spindle pole; ZP—zona pellucida; arrowheads indicate MTs of the spindle (only small part of each MT is shown). Scale bars: (**a**)—20 µm; (**b**,**c**)—5 µm; (**d**)—0.5 µm; (**e**)—1 µm.

**Figure 6 cells-12-01335-f006:**
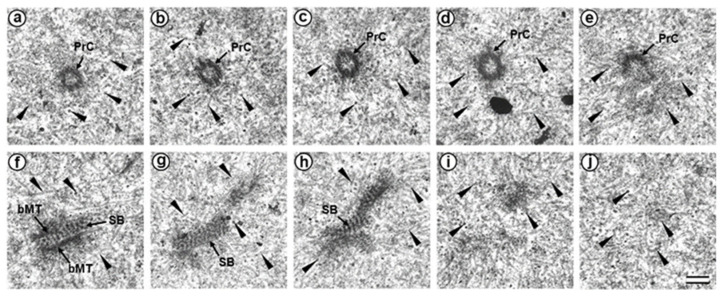
The flagellum-associated pole contained one centriole and a striated body in the prometaphase zygote of the first cleavage division (30 h after fertilization, serial sections from flagellum-associated pole). (**a**–**e**) serial sections of centriole; (**f**–**i**) serial sections of striated body. bMT—bundle of MT; PrC—proximal centriole; SB—striated body; arrowheads indicate MTs of the spindle (only small part of each MT is shown). Scale bar: 0.2 µm.

**Figure 7 cells-12-01335-f007:**
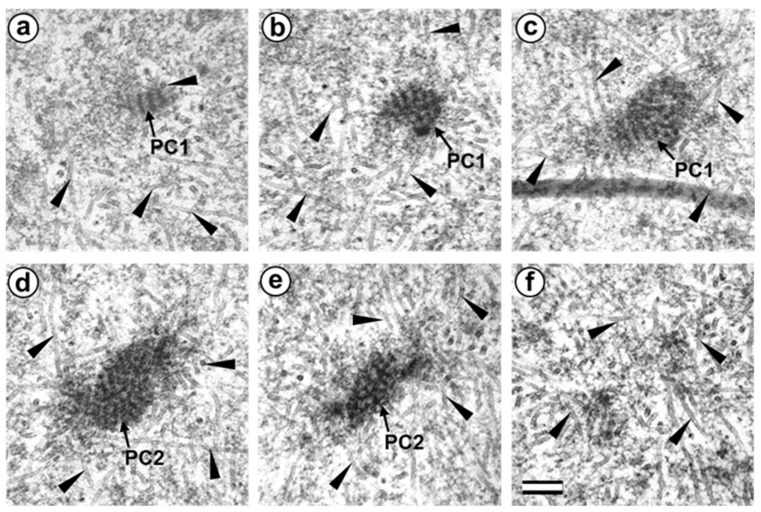
Two polar corpuscles are present in the pole opposite the flagellum-associated pole in the prometaphase zygote of the first cleavage division (30 h after fertilization, serial sections from opposite to flagellum-associated pole). (**a**–**c**) serial sections of polar corpuscle 1 (PC1); (**d**–**f**) serial sections of polar corpuscle 2 (PC2); arrowheads indicate MTs of spindle (only small part of each MT is shown). Scale bar: 0.2 µm.

**Figure 8 cells-12-01335-f008:**
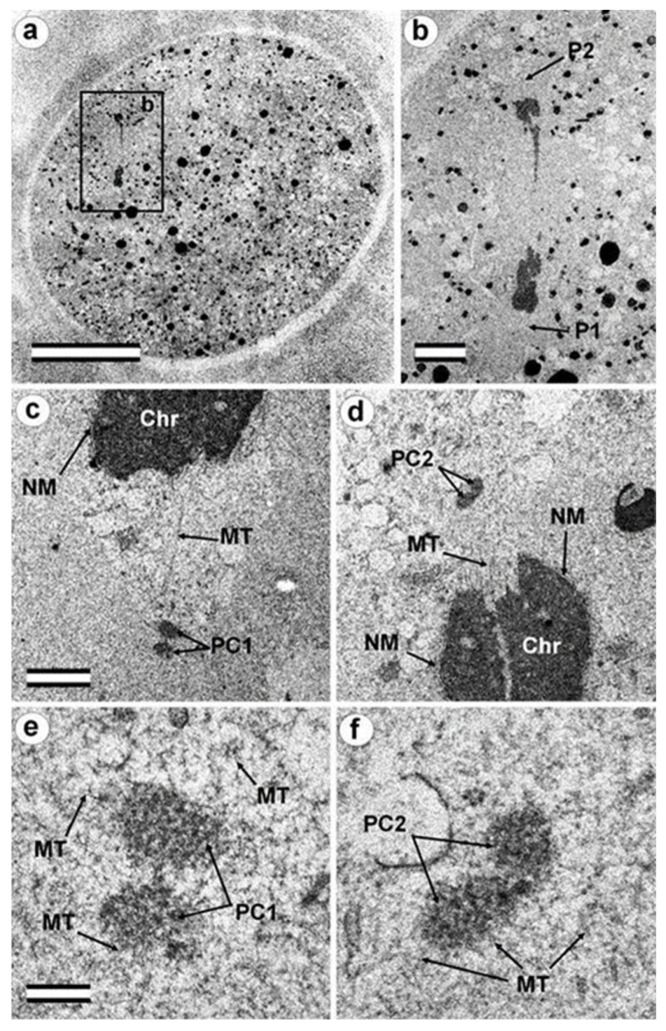
The late anaphase–early telophase zygote has two polar corpuscles in each pole; one of them associated with the apical part of the flagellar axoneme (30 h after fertilization). (**a**) General view of the embryo at low magnification; (**b**) the mitotic spindle area; (**c**) bottom pole region (P1); (**d**) top pole region (P2); (**e**) polar corpuscles in pole P1; (**f**) polar corpuscles in pole P2. Chr—chromosomes; MT—microtubules; NM—nuclear membrane; P1—bottom mitotic pole No. 1; PC1—polar corpuscles in mitotic pole No. 1; P2—top mitotic pole No. 2; PC2—polar corpuscles in mitotic pole No. 2. Scale bars: (**a**)—20 µm; (**b**)—5 µm; (**c**,**d**)—1 µm; (**e**,**f**)—0.2 µm. All 20 high-magnification serial sections of the bottom pole of the spindle and apical end of the flagellum are presented in Appendix A. 10 high-magnification serial sections of the top pole of the spindle are presented in Appendix A.

**Figure 9 cells-12-01335-f009:**
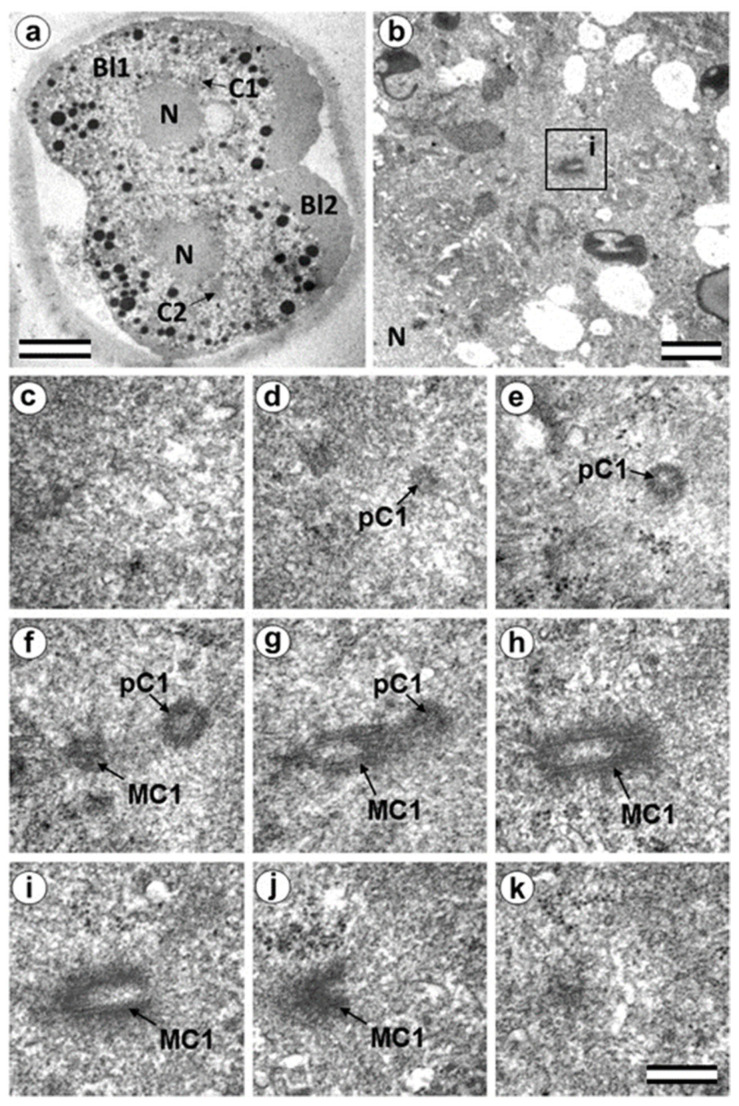
In two-cell embryo I, Blastomere 1 had two centrioles (30 h after fertilization). (**a**) General view of the embryo at low magnification; (**b**) the centrosomal region of blastomere Bl1; (**c**–**k**) serial sections through the centrosome of blastomere Bl1 at high magnification. Bl1—blastomere No. 1; Bl2—blastomere No. 2; C1—centrosome No. 1; C2—centrosome No. 2; MC1—mother centriole of centrosome No. 1; N—nucleus; pC1—procentriole of centrosome No. 1; ZP—Zona pellucida. Scale bars: (**a**)—20 µm; (**b**)—1 µm; (**c**–**k**)—0.3 µm.

**Figure 10 cells-12-01335-f010:**
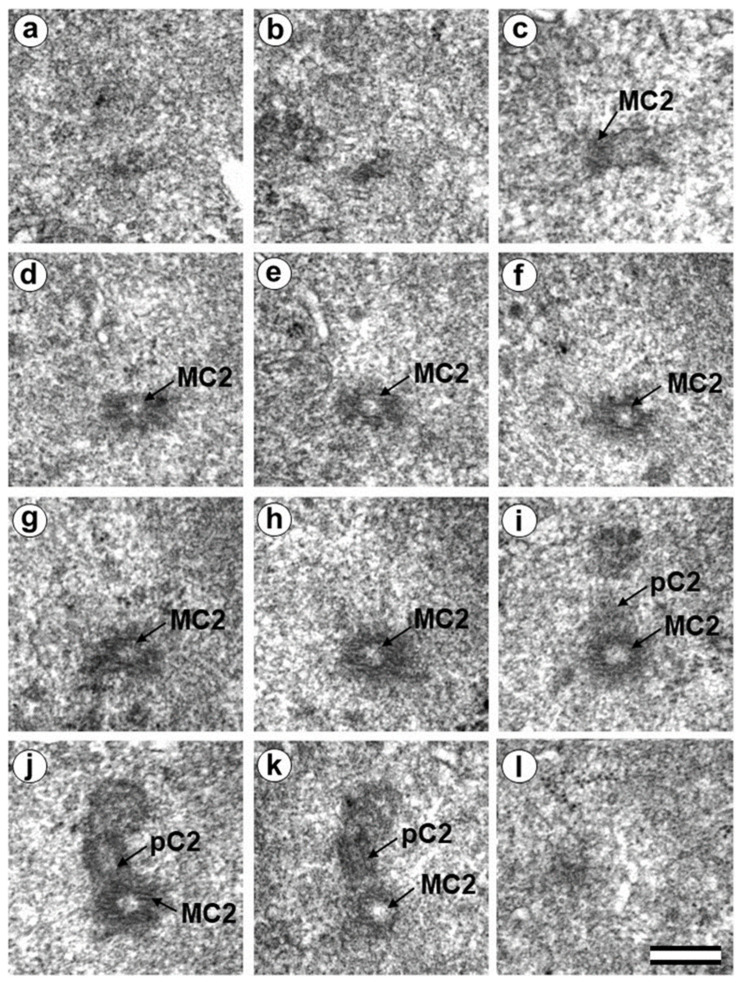
In two-cell embryo I, Blastomere 2 had two typical centrioles (30 h after fertilization). (**a**–**l**) Serial sections through the centrosome of blastomere Bl2 at high magnification; MC2—mother centriole of centrosome No. 2; pC2—procentriole of centrosome No. 2. Scale bar: 0.3 µm.

**Figure 11 cells-12-01335-f011:**
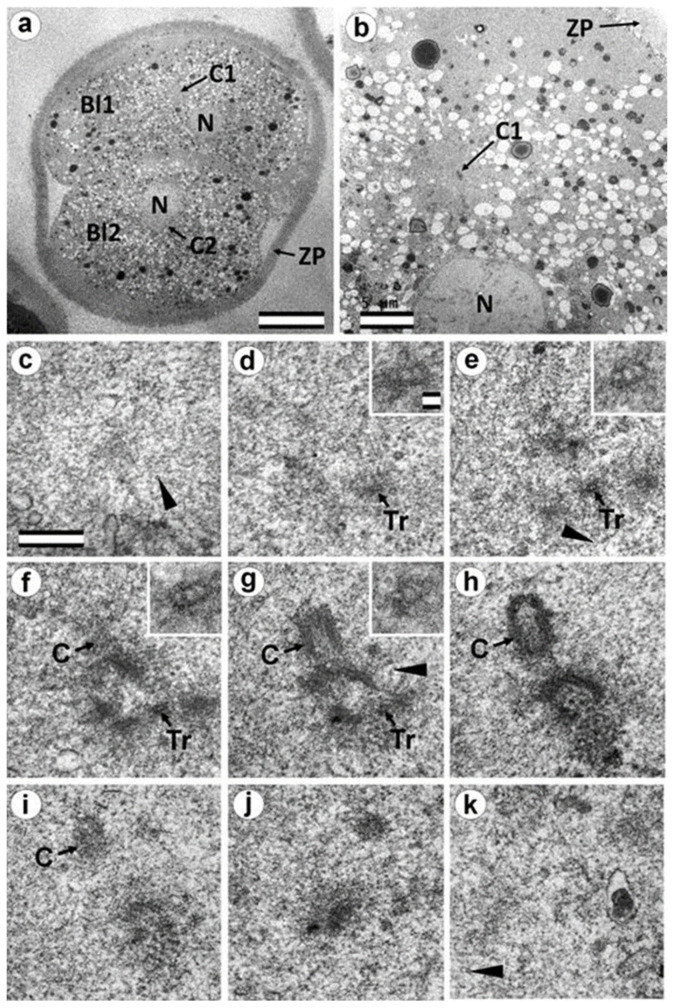
In two-cell embryo II, Blastomere 1 had 1 typical centriole and one atypical centriole (30 h after fertilization). (**a**) General view of the embryo at low magnification; (**b**) the centrosomal area of blastomere Bl1; (**c**–**j**) serial sections through the centrosome of blastomere Bl1 at high magnification; (**k)** one section after section j without centrosome. Bl1—blastomere No. 1; Bl2—blastomere No. 2; C1—centrosome No. 1; C2—centrosome No. 2; N—nucleus; Tr—triplets of microtubules; ZP—Zona pellucida; arrowheads indicate MTs in cytoplasm. Scale bars: (**a**)—20 µm; (**b**)—5 µm; (**c**–**k**)—0.3 µm; inserts—25 nm.

**Figure 12 cells-12-01335-f012:**
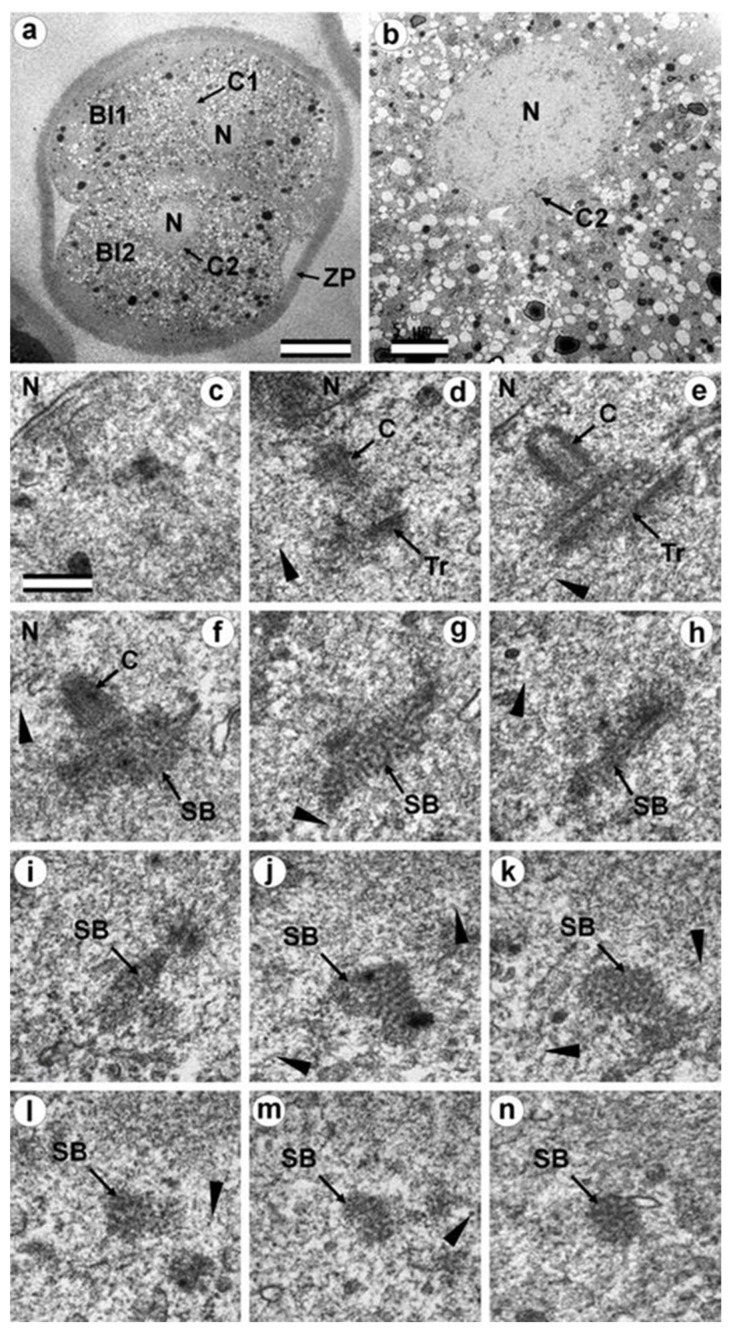
In two-cell embryo II, Blastomere 2 had one almost-typical centriole and one atypical centriole (30 h after fertilization). (**a**) General view of the embryo at low magnification; (**b**) the centrosomal area of blastomere Bl2; (**c**–**n**) serial sections through the centrosome of blastomere Bl2 at high magnification. Bl1—blastomere No. 1; Bl2—blastomere No. 2; C1—centrosome No. 1; C2—centrosome No. 2; N—nucleus; SB—striated body; Tr—triplets of microtubules; ZP—Zona pellucida; arrowheads indicate MTs in cytoplasm. Scale bars: (**a**)—20 µm; (**b**)—5 µm; (**c**–**k**)—0.3 µm.

**Figure 13 cells-12-01335-f013:**
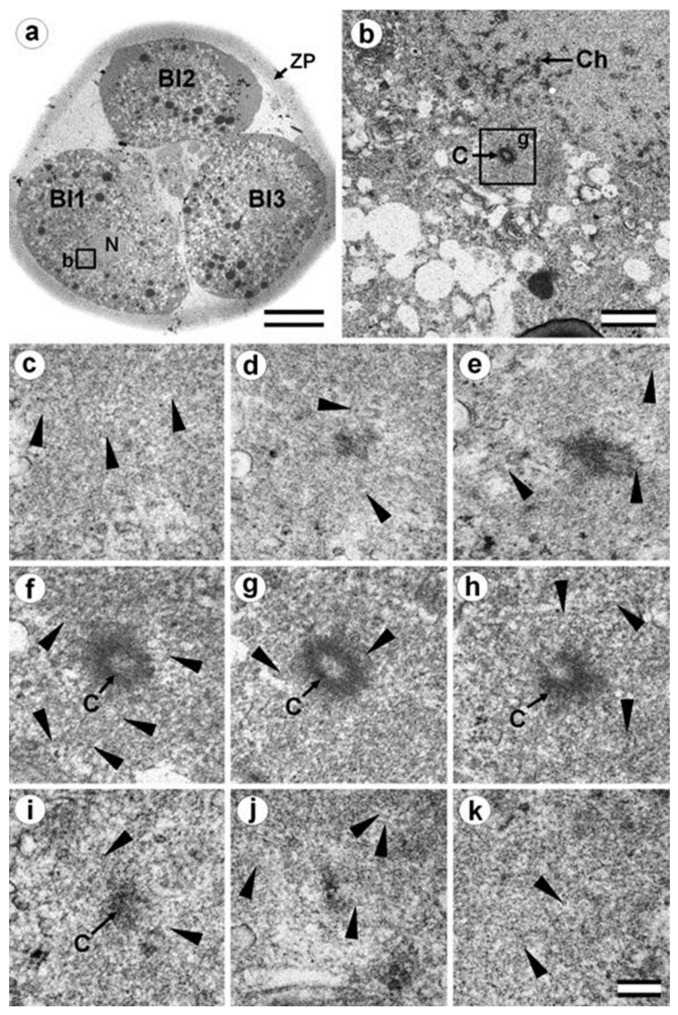
Blastomeres of the three-cell embryo whose centrosomal triplets were associated with electron-dense material (30 h after fertilization, prophase of the second cleavage division). (**a**) General view of the embryo at low magnification; (**b**) the centrosomal region of blastomere Bl1, left pole; (**c**–**k**) serial sections through left pole of blastomere Bl1 at high magnification. Bl1—blastomere No. 1; Bl2—blastomere No. 2; Bl3—blastomere No. 3; Ch—prophase chromosomes; ZP—zona pellucida; arrowheads indicate MTs in cytoplasm. Scale bars: (**a**)—20 µm; (**b**)—1 µm; (**c**–**k**)—0.2 µm.

**Figure 14 cells-12-01335-f014:**
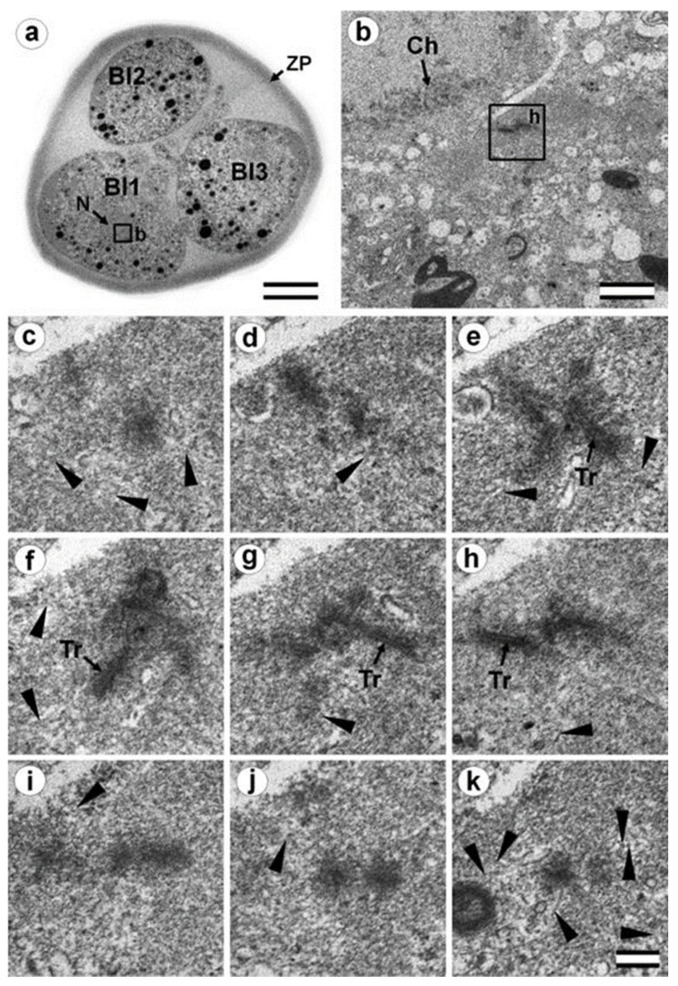
Blastomeres of the three-cell embryo with a centrosome containing several irregularly oriented short MT bundles (30 h after fertilization, prophase of the second cleavage division). (**a**) General view of the embryo at low magnification; (**b**) the centrosomal region of blastomere Bl1, right pole; (**c**–**k**) serial sections through right pole of blastomere Bl1 at high magnification. Bl1—blastomere No. 1; Bl2—blastomere No. 2; Bl3—blastomere No. 3; Ch—prophase chromosomes; N—nucleus; Tr—triplets of microtubules; ZP—zona pellucida; arrowheads indicate MTs in cytoplasm. Scale bars: (**a**)—20 µm; (**b**)—1 µm; (**c**–**k**)—0.2 µm. Sections before and after the series are shown in Appendix A.

**Figure 15 cells-12-01335-f015:**
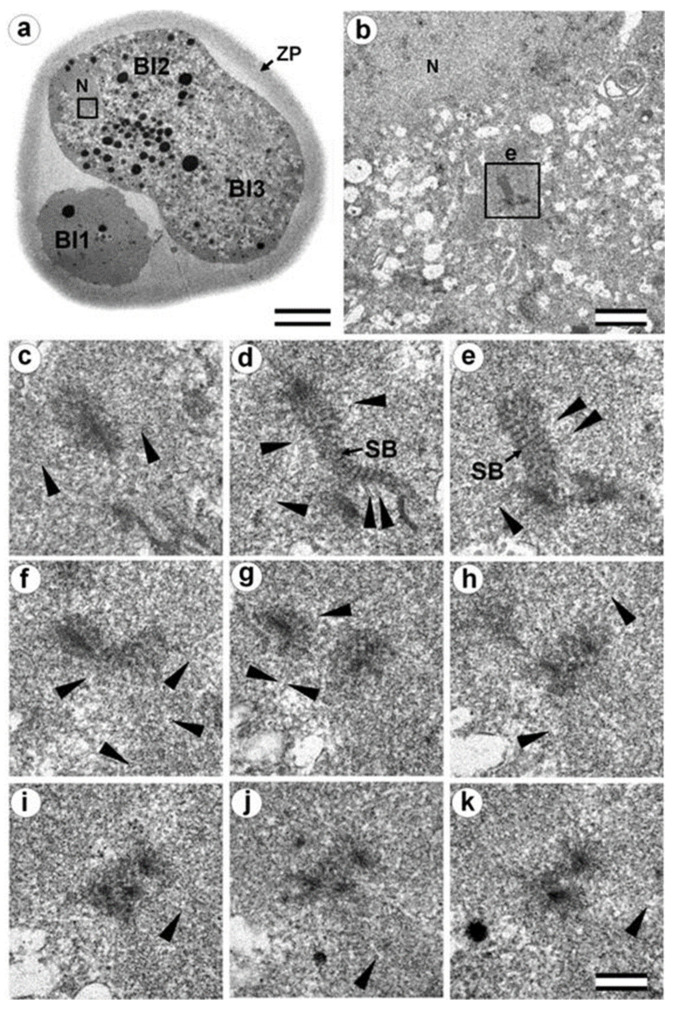
Cytotomy blastomeres of the three-cell embryo with centrosomes containing irregularly directed MT bundles (30 h after fertilization, cytotomy after the second cleavage division). (**a**) General view of the embryo at low magnification; (**b**) the centrosomal region of blastomere Bl2; (**c**–**k**) serial sections through the centrosome of blastomere Bl2 at high magnification. Bl1—blastomere No. 1; Bl2—blastomere No. 2; Bl3—blastomere No. 3; N—nucleus; SP—striated body; ZP—zona pellucida; arrowheads indicate MTs in cytoplasm. Scale bars: (**a**)—20 µm; (**b**)—5 µm; (**c**–**k**)—0.2 µm. Sections before and after the series are shown in Appendix A.

**Figure 16 cells-12-01335-f016:**
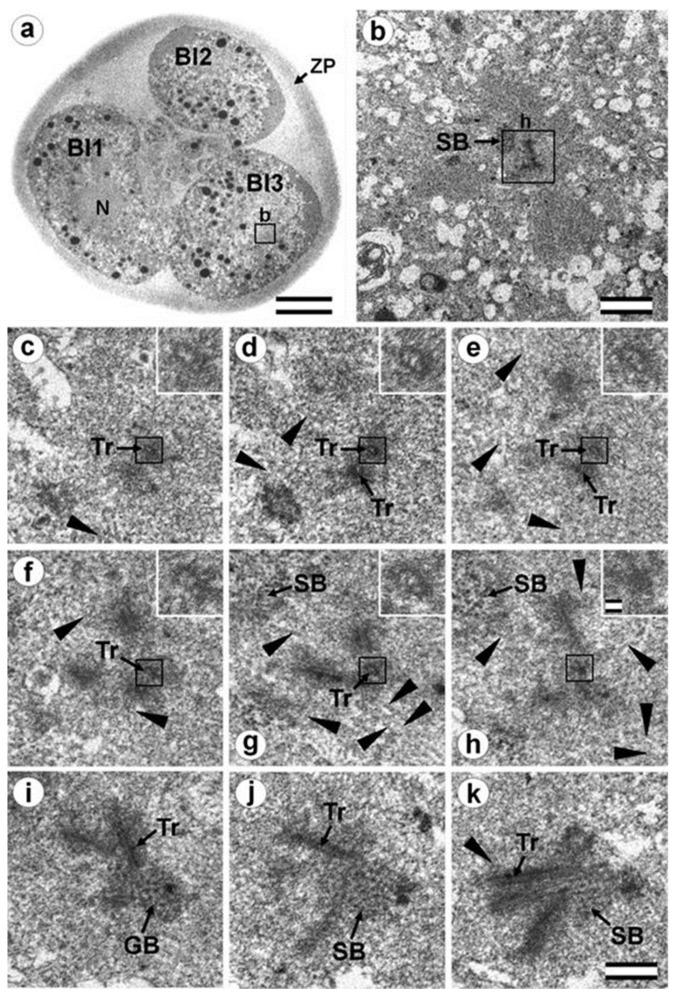
Cytotomy blastomeres of the three-cell embryo with centrosomes containing irregularly directed MT bundles (30 h after fertilization, cytotomy after the second cleavage division). (**a**) General view of the embryo at low magnification; (**b**) the centrosomal area of blastomere Bl3; (**c**–**k**) serial sections through the centrosome of blastomere Bl3 at high magnification. Bl1—blastomere No. 1; Bl2—blastomere No. 2; Bl3—blastomere No. 3; N—nucleus; SB—striated body; Tr—separated triplets of MT; ZP—zona pellucida; arrowheads indicate MTs in cytoplasm. Scale bar: (**a**)—20 µm; (**b**)—5 µm; (**c**–**k**)—0.2 µm; inserts—25 nm. Sections before and after the series are shown in Appendix A.

**Figure 17 cells-12-01335-f017:**
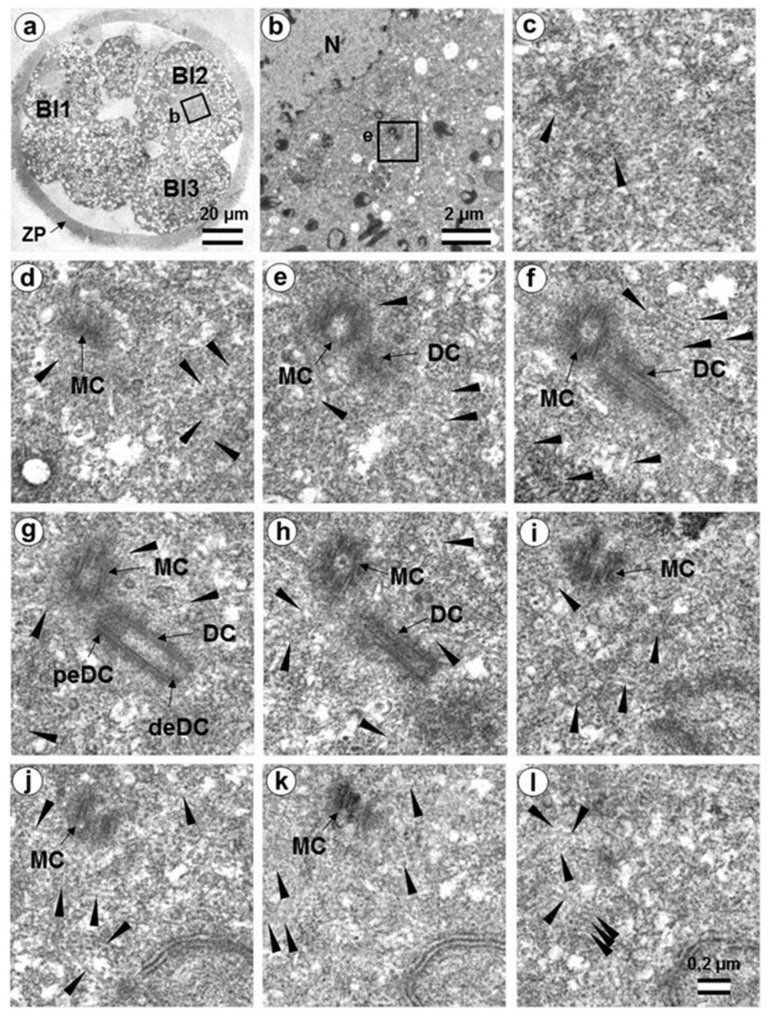
The flagellum-free blastomere Bl2 of a 14-cell embryo had a classical centrosome with two long, typical centrioles (36 h after fertilization, early interphase after the third division of cleavage). (**a**) General view of the embryo at low magnification; (**b**) the centrosomal area of blastomere Bl2; (**c**–**l**) serial sections through the centrosome of blastomere Bl2 at high magnification. Bl1—blastomere No. 1; Bl2—blastomere No. 2; Bl3—blastomere No. 3; DC—daughter centriole; deDC—distal end of daughter centriole; peDC—proximal end of daughter centriole; MC—mother centriole; N—nucleus; ZP—zona pellucida; arrowheads indicate MTs in cytoplasm. Scale bars: (**a**)—20 µm; (**b**)—2 µm; (**c**–**k**)—0.2 µm.

**Figure 18 cells-12-01335-f018:**
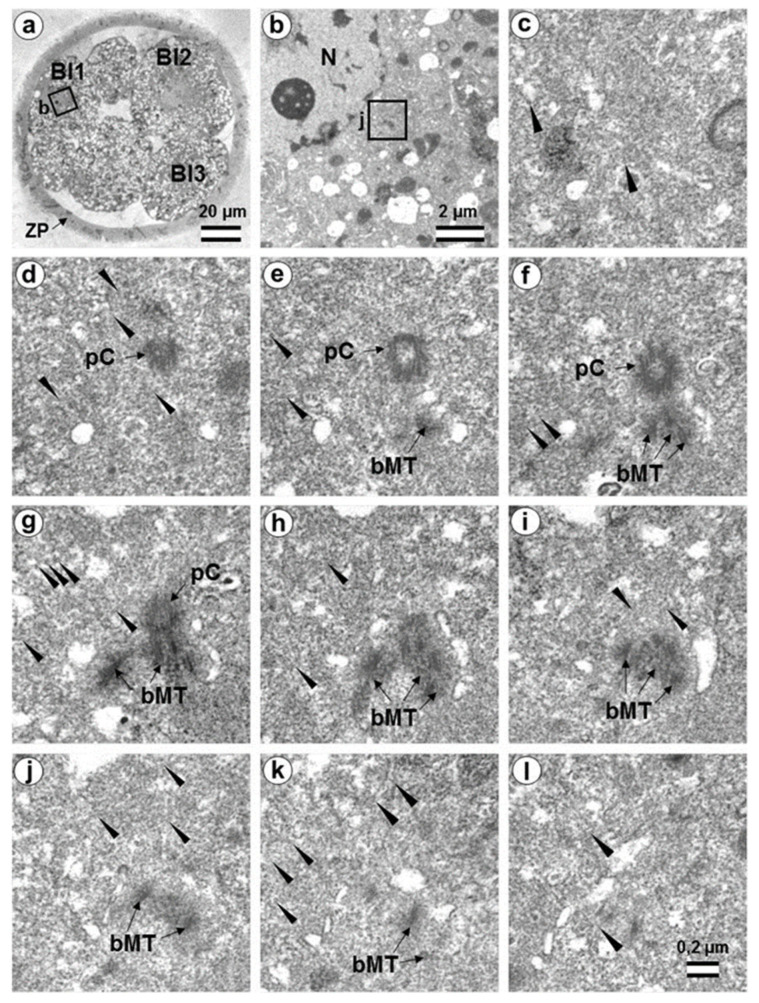
The first centrosome in flagella-associated blastomere Bl1 of a 14-cell embryo has an atypical centriole and a small typical procentriole (36 h after fertilization). (**a**) General view of the embryo at low magnification; (**b**) the centrosomal area of blastomere Bl1; (**c**–**l**) 10 serial sections through the centrosome of blastomere Bl1 at high magnification. Bl1—blastomere No. 1; Bl2—blastomere No. 2; Bl3—blastomere No. 3; bMT—bundles of MT of atypical centriole; N—nucleus; pC—procentriole; ZP—zona pellucida; arrowheads indicate MTs in cytoplasm. Scale bars: (**a**)—20 µm; (**b**)—2 µm; (**c**–**k**)—0.2 µm. Six subsequent serial sections containing the sperm flagellum axoneme are shown in Appendix A.

**Figure 19 cells-12-01335-f019:**
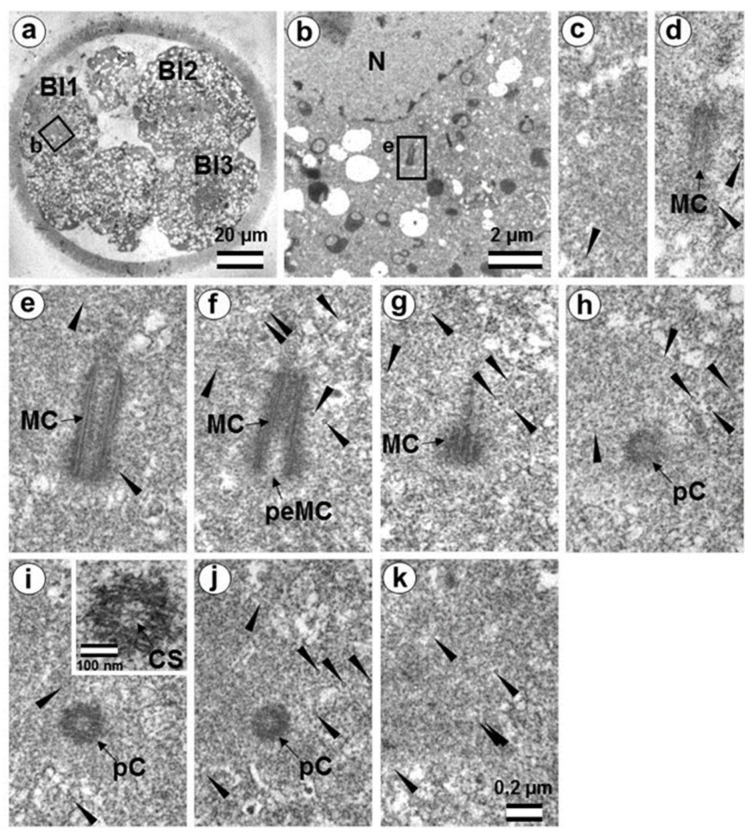
The second centrosome in flagella-associated blastomere Bl1 of a 14-cell embryo has a classical centrosome with a typical mother centriole and a small typical procentriole (36 h after fertilization, late interphase after the third cleavage division). (**a**) General view of the embryo at low magnification; (**b**) the centrosomal area of blastomere Bl1; (**c**–**k**) serial sections through the centrosome of blastomere Bl1 at high magnification. Bl1—blastomere No. 1; Bl2—blastomere No. 2; Bl3—blastomere No. 3; CS—cartwheel structure; MC—mother centriole; N—nucleus; pC—procentriole; peMC—proximal end of mother centriole; ZP—zona pellucida; arrowheads indicate MTs in cytoplasm. Scale bars: (**a**)—20 µm; (**b**)—2 µm; (**c**–**k**)—0.2 µm; inserts—100 nm.

**Figure 20 cells-12-01335-f020:**
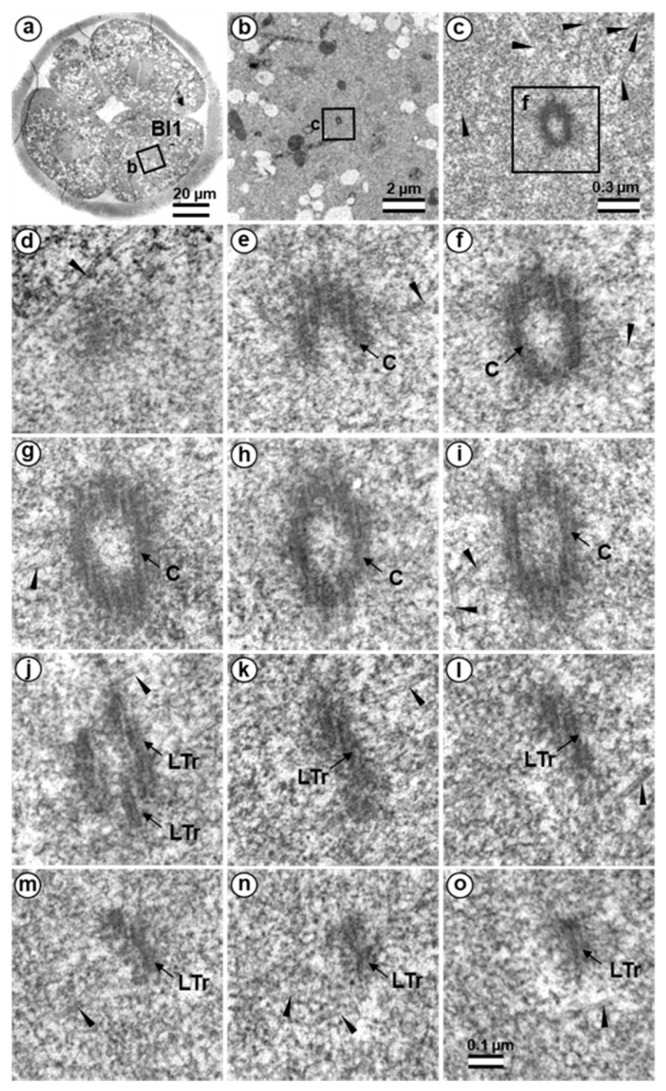
The centriole in four-cell parthenogenetic embryo 2. (**a**) General view of the embryo at low magnification; (**b**) the centrosomal area of blastomere Bl1; (**c**) centriole at middle magnification; (**d**–**o**) serial sections through the centriole of blastomere Bl1 at high magnification. Bl1—blastomere No. 1; C—centriole; LTr—long triplets of centriole; arrowheads indicate MTs in cytoplasm. Scale bars: (**a**)—20 µm; (**b**)—2 µm; (**c**)—0.3 µm; (**d**–**o**)—100 nm.

**Figure 21 cells-12-01335-f021:**
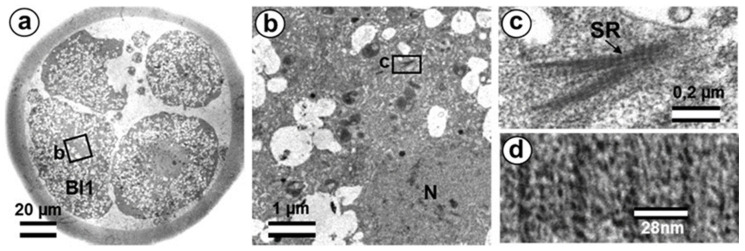
The centriole in four-cell parthenogenetic embryo 1. (**a**) General view of the embryo at low magnification; (**b**) the rootlet area of blastomere Bl1; (**c**) rootlet at high magnification; (**d**) distance between lines of striated rootlet. Bl1—blastomere No. 1; SR—striated rootlet. Scale bars: (**a**)—20 µm; (**b**)—1 µm; (**c**)—0.2 µm; (**d**)—28 nm.

**Figure 22 cells-12-01335-f022:**
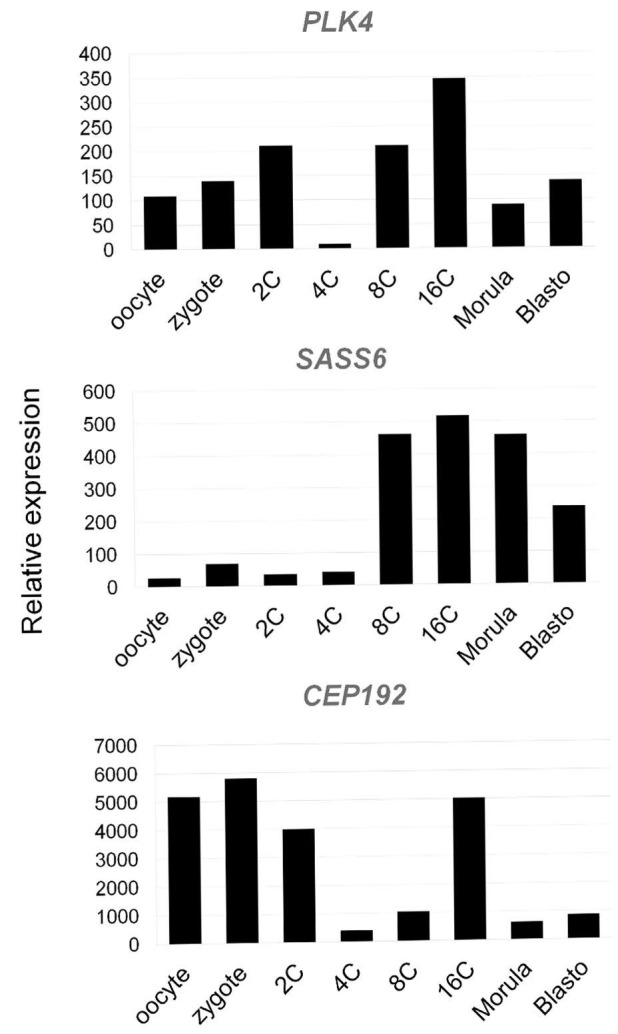
Expression of genes coding Polo-Like Kinase 4 (PLK4), SAS-6 Centriolar Assembly Protein (SASS6), and Centrosomal Protein of 192 kDa (CEP192) in bovine oocytes and preimplantation embryos using Affymetrix microarray (GEO accession GDS3960, Preimplantation embryonic development in bovine). Mean expression values of genes PLK4, SASS6, and CEP192 in oocyte, zygote, 2-cell (2C), 4-cell (4C), 8-cell (8C), 16-cell (16C), morula, and blastocyst stages are presented.

**Figure 23 cells-12-01335-f023:**
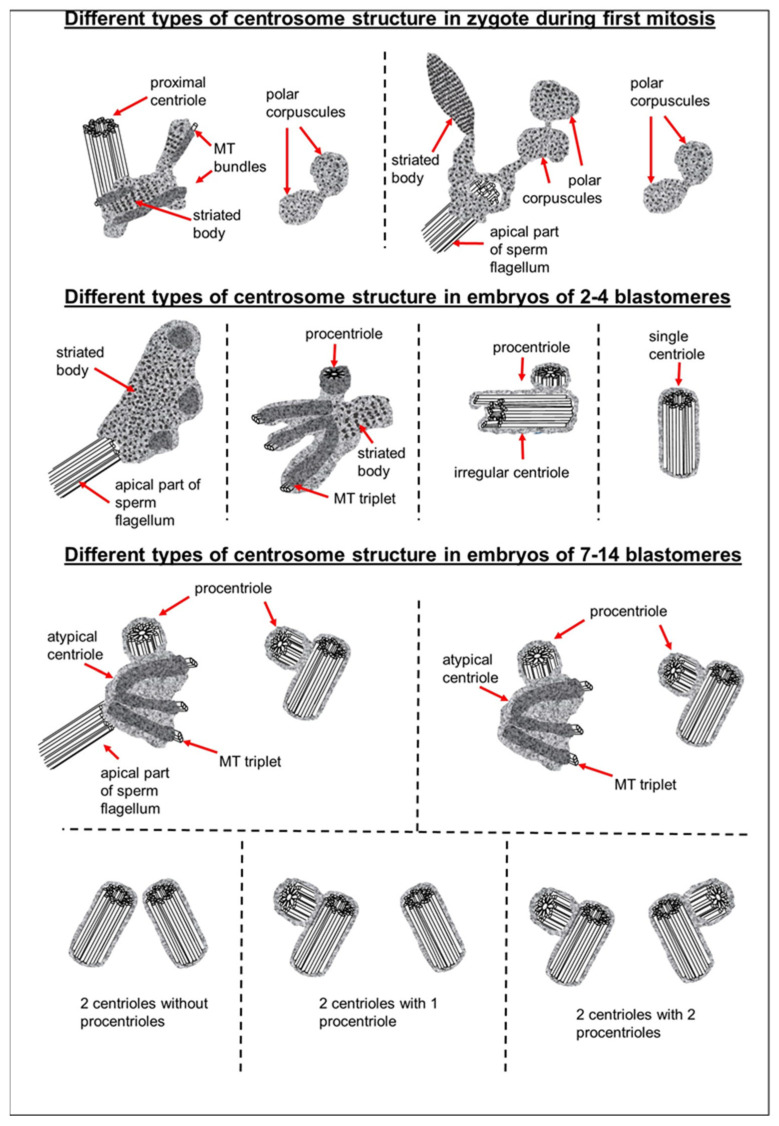
Centrosomal structure during the early stages of bovine embryonic development.

**Figure 24 cells-12-01335-f024:**
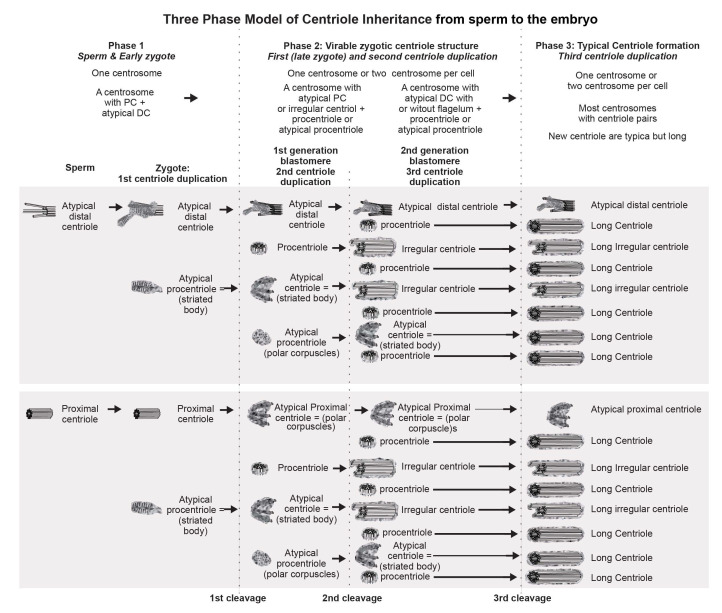
Formation of canonical centrioles during early embryonic development in cattle.

**Table 1 cells-12-01335-t001:** Lengths of centrioles and procentrioles in different blastomeres of a seven-cell embryo.

Blastomere	Mother Centriole (Length in nm)	Procentriole of Mother Centriole (Length in nm)	Daughter Centriole (Length in nm)	Procentriole of Daughter Centriole (Length in nm)
Cell 1	700	333	623	407
Cell 2	652	488	584	357
Cell 3 (flagellum in cytoplasm)	Atypical centriole	312	607	280
Cell 4	812	405	593	378
Cell 5	1059	518	607	280
Cell 6	732	292	574	312
Cell 7	Atypical centriole	474	754	311
**Average**	*** 791 ± 161**	**403 ± 92**	**620 ± 61**	**332 ± 49**

Note: The sizes of centrioles and procentrioles were calculated as the ratio of their length to the scale bar. See the Materials and Methods section for a method for calculating the length on oblique sections. * Without atypical centrioles.

**Table 2 cells-12-01335-t002:** Lengths of centrioles and procentrioles in blastomeres of a 14-cell embryo.

Blastomere	Mother Centriole (Length in nm)	Procentriole of Mother Centriole (Length in nm)	Daughter Centriole (Length in nm)	Procentriole of Daughter Centriole (Length in nm)
Cell 1 (flagellum in cytoplasm)	Atypical centriole	366	644	194
Cell 2	786	No procentriole	645	No procentriole
Cell 3	870	No procentriole	755	No procentriole
Cell 4	1024	443	640	410
Cell 5	984	397	797	138
Cell 6	814	No procentriole	644	No procentriole
Cell 7	Atypical centriole	394	735	259
Cell 8	892	No procentriole	625	No procentriole
Cell 9	778	No procentriole	740	No procentriole
Cell 10	848	319	691	163
Cell 11	897	No procentriole	644	No procentriole
Cell 12	729	452	724	392
Cell 13	897	238	677	148
Cell 14	844	350	666	No procentriole
**Average**	*** 864 ± 84**	**370 ± 69 (** 211 ± 197)**	**688 ± 53**	**243 ± 115 (** 122 ± 148)**

**Note:** The sizes of centrioles and procentrioles were calculated as the ratio of their length to the scale bar. See the Materials and Methods section for a method for calculating the length on oblique sections. * Without atypical centrioles. ** Calculation includes blastomeres without procentrioles (procentriole length = 0).

## Data Availability

Not applicable.

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
