# Peer review of "Centrosome Formation in the Bovine Early Embryo"

_cells, 2023, doi:10.3390/cells12091335_

Round 1

Reviewer 1 Report

The work of Uzbekov et al. is devoted to an important and urgent problem - the problem of inheritance of centrioles in the early development of animals. Are centrioles a relatively independent, inherited organelle, or are they formed de novo? Uzbekov et al. studied  centrioles in zygotes and early cattle embryos, including parthenogenetic embryos. The study was done in the most thorough manner, on serial ultrathin sections (1200-1500 sections in a row!). The material collected authors is presented and illustrated in the proposed article. I believe that this interesting article should certainly be published. Meanwhile, its text needs some editing.

1) The introduction should be condensed and clearly state the purpose.

2) The discussion should also be shortened, and one wants to see the conclusions or assumptions made by the author in it. Perhaps the conclusion will be that centrioles are formed de novo, with some randomness in different blastomeres, and in the process of their formation they undergo phases of unbalanced morphogenesis.

3) It is desirable to provide each illustration with a serially cut centrosome with a schematic representation of this centrosome.

4) Tables with centriole lengths should be moved to the supplement, leaving the average length in the main text.

5) The authors would do well to explain in the text how they distinguish between maternal and daughter centrioles and centrioles in their images.

6) The data on the expression of centrosome proteins in embryos are very interesting. However, it is necessary to explain in more detail where these data come from.

Author Response

Dear Editor!

Firstly, I would like to thank all the reviewers for their detailed analysis of our work. I myself review a lot of articles and I understand that reviewing a such big paper is hard work. Below I give answers and explanations on all points. Necessary changes made to text are marked in red. The answers also include line numbers for a more accurate search for additions and references made to the text of the article. 

Reviewer 1

The work of Uzbekov et al. is devoted to an important and urgent problem - the problem of inheritance of centrioles in the early development of animals. Are centrioles a relatively independent, inherited organelle, or are they formed de novo? Uzbekov et al. studied centrioles in zygotes and early cattle embryos, including parthenogenetic embryos. The study was done in the most thorough manner, on serial ultrathin sections (1200-1500 sections in a row!). The material collected authors is presented and illustrated in the proposed article. I believe that this interesting article should certainly be published. Meanwhile, its text needs some editing.

Comment 1.

 The introduction should be condensed and clearly state the purpose.

Answer 1.  Thanks for these comments. We agree that the introduction is long and we deleted the last paragraph. However, we felt the reset of the introduction is needed because different studies on this issue have reported very conflicting results and therefore we would like presenting different points of view and thus justify the objectives of our work.

The objective was clearly mentioned (lines 104-106), but since it was not sufficiently apparent, we have modified it and expanded on it.

Comment 2.

 The discussion should also be shortened, and one wants to see the conclusions or assumptions made by the author in it. Perhaps the conclusion will be that centrioles are formed de novo, with some randomness in different blastomeres, and in the process of their formation they undergo phases of unbalanced morphogenesis.

Answer 2. Thanks for these comments. Unfortunately, we cannot yet unequivocally state that the proximal centriole of the spermatozoon is ALWAYS resorbed during the first division and all subsequent centrioles are formed exclusively de novo. According to our data, we can state that the proximal centriole MAY be resorbed and the first cleavage division MAY take place without the participation of centrioles. The morphology of centrioles in the two-cell embryos that we studied did not correspond to the morphology of the proximal centriole of the spermatozoon, but the transformation of the centriole morphology during the transition from the zygote to the two-cell embryo cannot be ruled out. We agree with the reviewer's definition that the process of centriole formation goes through the stage of "unbalanced morphogenesis". We have sufficiently detailed it in the figures in the article. And yet, for the time being, we prefer to be cautious in making an unambiguous conclusion about completely newly formed centrioles in early cow embryos, taking into account the data of other researchers. Now we continue our studies using correlation microscopy, combining electron microscopy on serial sections and immunocytochemical examination of various centrosomal proteins in early bovine embryos using super-resolution light microscopy.

Thanks for these comments and we added to Discussion sentence about “unbalanced morphogenesis of centriole” (line 692) and also to conclusions (lines 887-888).

Comment 3.

 It is desirable to provide each illustration with a serially cut centrosome with a schematic representation of this centrosome.

Answer 3. Thanks for these comments. In the paper, we have presented serial sections of centrosomes at all stages from the zygote to the 14-cell embryo. For readers who are not experts in electron microscopy, we have supplied the figures with pictures of centrosomes. Some of them are presented in the article next to the photographs (Fig. 2 and 3), some are presented in a separate figure 23 in the discussion for the convenience of comparing the morphology of centrosomes at different stages of development.

Comment 4.

 Tables with centriole lengths should be moved to the supplement, leaving the average length in the main text.

Answer 4. We cannot agree with this reviewer's proposal, since the average length of centrioles is only one of the results presented in the table. A complete description of the results of the tables would require a significant expansion of the descriptive text. In the editorial recommendations, authors are encouraged to use tables for a more visual presentation of this type of results.

Comment 5.

 The authors would do well to explain in the text how they distinguish between maternal and daughter centrioles and centrioles in their images.

Answer 5. This explanation was given in our article (lines 488-491):

«Closely oriented long centrioles formed the diplosome. The mother centriole could be distinguished due to mutual arrangement of centriolar cylinders. Both ends of mother centrioles are "free," i.e., not covered by the second centriole (Fig. 16), whereas one of the ends of the daughter centriole is directed towards the surface of the mother centriole (Figure 17 g).»

Comment 6.

The data on the expression of centrosome proteins in embryos are very interesting. However, it is necessary to explain in more detail where these data come from.

Answer 6. Thanks for these comments. The details were added in Material and Method section (Lines 168-173), and also are described in the legend of Fig. 22.

«2.4. Gene expression

Transcriptome data on bovine preimplantation embryonic development were retrieved from the NCBI Gene Expression Omnibus database (GEO, https://www.ncbi.nlm.nih.gov/geo/; GEO dataset accession GDS3960). Mean expression values of genes PLK4, SASS6, and CEP192 were calculated from the values reported from two independent microarray hybridizations of mRNA from bovine oocytes, zygotes, 2-cell (2C), 4-cell (4C), 8-cell (8C), 16-cell (16C), morula, and blastocysts. Gene expression values at different stage of embryo development were presented as a histogram. »

Reviewer 2 Report

The illustrative material of the article is exhaustive, however, I have a several minor remarks regarding figure legends.

In Figure 10 legend we can read "LC2 – long centriole of centrosome No. 2; SC2 – short centriole of centrosome 2", but at the corresponding panels there are "pC2" and "MC2" (presumably, procentriole of centrosome No. 2 and mother centriole of centrosome No. 2).

In Figure 12 legend there are no definitions for "SB" and "Tr" on high magnification panels (presumably, striated body and triplets of microtubules)

In Figure 14 legend: the same as in figure 12 - no definition for the "Tr"

In Figure 16: misprint in legend - SP instead of SB correspond to "striated body"

Also, perhaps it should touch at least briefly in the Discussion on the very interesting phenomenon of the sharp drop in the expression level of PLK4 at the stage of the 4-cell embryo, shown on figure 22

Author Response

Dear Editor!

Firstly, I would like to thank all the reviewers for their detailed analysis of our work. I myself review a lot of articles and I understand that reviewing a such big paper is hard work. Below I give answers and explanations on all points. Necessary changes made to text are marked in red. The answers also include line numbers for a more accurate search for additions and references made to the text of the article. 

Reviewer 2

The illustrative material of the article is exhaustive, however, I have a several minor remarks regarding figure legends.

Comment 1.

In Figure 10 legend we can read "LC2 – long centriole of centrosome No. 2; SC2 – short centriole of centrosome 2", but at the corresponding panels there are "pC2" and "MC2" (presumably, procentriole of centrosome No. 2 and mother centriole of centrosome No. 2).

Answer 1. Thanks for these comments. We have made the necessary corrections to the legend of Figure 10.

Comment 2.

In Figure 12 legend there are no definitions for "SB" and "Tr" on high magnification panels (presumably, striated body and triplets of microtubules)

Answer 2. Thanks for these comments. We have made the necessary corrections to the legend of Figure 12. 

Comment 3.

In Figure 14 legend: the same as in figure 12 - no definition for the "Tr"

Answer 3. Thanks for these comments. We have made the necessary corrections to the legend of Figure 14.

Comment 4.

In Figure 16: misprint in legend - SP instead of SB correspond to "striated body"

Answer 4. Thanks for these comments. We have made the necessary corrections to the legend of Figure 16.

Comment 5.

Also, perhaps it should touch at least briefly in the Discussion on the very interesting phenomenon of the sharp drop in the expression level of PLK4 at the stage of the 4-cell embryo, shown on figure 22

Answer 5. We added new paragraph in the text (Lines 688-692 in new version):

«According to gene expression patterns of SASS6, CEP192 and PLK4 during early embryo development, both CEP192 and PLK4 demonstrated degradation of maternal mRNA pool at 4C, and reactivation of transcription at 8C, when major genome activation starts in bovine embryo. Therefore, in 4-cell embryo there is likely a shortage of PLK4 and CEP192 transcripts that may lead to lack of corresponding proteins, and consequently, "unbalanced morphogenesis" of centrioles observed at stages 2 and 4 blastomeres. »

Reviewer 3 Report

The manuscript of Uzbekov et al. entitled “Centrosome formation in the bovine early embryo” aims to clarify the inheritance of paternal centrioles from spermatozoa and centrosome formation in the early bovine embryo. To this goal the AA made a careful ultrastructural analysis by giving hundred ultrathin serial sections through whole early embryos at different stages of development. This procedure is very difficult since many sections can be lost during the process, but the AA demonstrate an excellent mastery of the technique obtaining very informative images. The centriole inheritance during early mammalian development is poorly understood, thus the present manuscript could give more information to the general knowledge of this interesting process. I have only a few observations before ethe paper can be published.

First at all, a little thought on a term that in my opinion is generally misleading in the literature, but that the AA don't necessarily have to take into consideration for their manuscript. Parthenogenesis is a form of reproduction wherein the offspring develops from the female gamete without male contribution. Therefore, I will be cautious to define the aborted development of early unfertilized embryos as a parthenogenetic process.  The process that the authors described is the early development of unfertilized eggs, that do not develop until offspring but degenerate very early. I know that this term was usually commonly used to define this development interrupted in earlier stages. However, I think this is incorrect, better parthenogenetic-like development in my opinion.

Figs. 10-11-12-14-16. Some labels are not explained in the legend.

Fig.22 . It is unclear for me how this picture was generated unless showing details of the given analysis.

Discussion

l.613-614. Please explain how the reduction of the cell dimension can improve the regulation of centriolar proteins. Why regulation?

l. 716. The centrioles do not necessarily have to nucleate a ciliary or flagellar axoneme. Many cells have centrioles that do not assemble ciliary structures.

l.720. The node form during late gastrulation. By this time centrioles are already found in all cells.

l.726. Female germ cells In Drosophila have centrioles with triplets but do not form ciliary structures. On the contrary, Johnston organs have axonemes nucleated by centriole doublets.

l.734-735. Why decrease in cell size may facilitate molecular regulation?

l.753. How the AA observed the increase of Sas6?

Author Response

Dear Editor!

Firstly, I would like to thank all the reviewers for their detailed analysis of our work. I myself review a lot of articles and I understand that reviewing a such big paper is hard work. Below I give answers and explanations on all points. Necessary changes made to text are marked in red. The answers also include line numbers for a more accurate search for additions and references made to the text of the article. 

Reviewer 3

The manuscript of Uzbekov et al. entitled “Centrosome formation in the bovine early embryo” aims to clarify the inheritance of paternal centrioles from spermatozoa and centrosome formation in the early bovine embryo. To this goal the AA made a careful ultrastructural analysis by giving hundred ultrathin serial sections through whole early embryos at different stages of development. This procedure is very difficult since many sections can be lost during the process, but the AA demonstrate an excellent mastery of the technique obtaining very informative images. The centriole inheritance during early mammalian development is poorly understood, thus the present manuscript could give more information to the general knowledge of this interesting process. I have only a few observations before the paper can be published.

Comment 1 First at all, a little thought on a term that in my opinion is generally misleading in the literature, but that the AA don't necessarily have to take into consideration for their manuscript. Parthenogenesis is a form of reproduction wherein the offspring develops from the female gamete without male contribution. Therefore, I will be cautious to define the aborted development of early unfertilized embryos as a parthenogenetic process.  The process that the authors described is the early development of unfertilized eggs, that do not develop until offspring but degenerate very early. I know that this term was usually commonly used to define this development interrupted in earlier stages. However, I think this is incorrect, better parthenogenetic-like development in my opinion.

Answer 1.  We agree with the reviewer that in this context it would be more accurate to use the term "parthenogenetic development" instead of the term "parthenogenesis".

We have replaced (line 126)

“In vitro embryo production by parthenogenesis”

to

“Parthenogenetic development of embryos in vitro

Comment 2.

  Figs. 10-11-12-14-16. Some labels are not explained in the legend.

Answer 2. Thanks for these comments. We have made the necessary corrections to the legend of Figures 10-11-12-14-16.

Comment 3.

Fig. 22. It is unclear for me how this picture was generated unless showing details of the given analysis.

Answer 3.  We added the information for clear explanation the origin of these histograms.

The details were added in Material and Methods (Lines 168-173) in new version, as well as in Figure 22 legend.

«2.4. Gene expression

Transcriptome data on bovine preimplantation embryonic development were retrieved from the NCBI Gene Expression Omnibus database (GEO, https://www.ncbi.nlm.nih.gov/geo/; GEO dataset accession GDS3960). Mean expression values of genes PLK4, SASS6, and CEP192 were calculated from the values reported from two independent microarray hybridizations of mRNA from bovine oocytes, zygotes, 2-cell (2C), 4-cell (4C), 8-cell (8C), 16-cell (16C), morula, and blastocysts. Gene expression values at different stage of embryo development were presented as a histogram. »

Discussion

Comment 4.

l.613-614. Please explain how the reduction of the cell dimension can improve the regulation of centriolar proteins. Why regulation?

Answer 4.  May be another lines? l.613-614 (in old paper version) are Figure 21 legend.

In our opinion, the effective regulation of processes in the cell that require the interaction of several proteins depends on two principal parameters: 1) the activity of the synthesis of regulatory proteins; 2) transport of these proteins from the site of synthesis to the site of their interaction. Obviously, with a decrease in the total volume of the cell, the required protein concentration is reached faster. Similarly, for transport, regardless of whether it is simple diffusion along a concentration gradient or any of the variants of active transport, the distance from the site of synthesis (or the volume of the cell during diffusion) is greater in a large cell. Cell volume can also be important in the degradation of proteins during the transition from one functional state of the cell to another.

Comment 5.

  1. 716. The centrioles do not necessarily have to nucleate a ciliary or flagellar axoneme. Many cells have centrioles that do not assemble ciliary structures.

Answer 5.

It certainly is. There are even cell lines whose cells never form primary cilia, let alone mobile cilia or flagella. However, in the process of embryonic development, the presence of a cilium (the possibility of its formation by certain cells) is of fundamental importance, in particular, for the laying of right-left symmetry.

Comment 6.

l.720. The node form during late gastrulation. By these time centrioles are already found in all cells.

Answer 6.

Strictly speaking, for such a statement it is necessary to examine all the cells of the embryo. Probably, although this statement requires a special study, atypical centrioles remain in two cells even at this stage (and in one of the cells, the remnants of the axoneme of the spermatozoon flagellum and, possibly, even partially the mitochondria of the spermatozoon).

Comment 7.

l.726. Female germ cells In Drosophila have centrioles with triplets but do not form ciliary structures. On the contrary, Johnston organs have axonemes nucleated by centriole doublets.

Answer 7.

It is certainly true that the presence of triplets is not necessarily directly related to cilia production. At least at this stage of development. The absence of cilia in Drosophila germ cells was specifically shown by electron microscopy in an article by Riparbelli et al., 2021. Cells; 10(8): 1997. However, in germ cells of male Drosophila, centrioles containing MT triplets form primary cilia or their analogs.

As for the sense organ (I know a beautiful work on this topic Persico et al. 2019. Front Cell Dev Biol. 2019; 7: 173. ) located in insect antennae (Johnston organ), this example shows that at least in insects MT triplets are not necessary for the successful growth of the primary cilium or its analogue.

In general, insects, due to their evolutionary antiquity, huge numbers and reproduction rates, have accumulated a lot of deviations from the structure of the classical centrosome. In particular, in my article, which is currently under review, the centriole wall in neurons of wasps Anisopteromalus calandrae female cells consists of microtubule singlets, and the wall of the basal body of male spematocytes of Anisopteromalus calandrae consists of MT triplets, and the larval centriole does not contain MT at all (Uzbekov et al., 2018, Open Biology, 7, bio036012).

 Comment 8.

l.734-735. Why decrease in cell size may facilitate molecular regulation?

Answer 8 =  Answer 4 In our opinion, the effective regulation of processes in the cell that require the interaction of several proteins depends on two principal parameters: 1) the activity of the synthesis of regulatory proteins; 2) transport of these proteins from the site of synthesis to the site of their interaction. Obviously, with a decrease in the total volume of the cell, the required protein concentration is reached faster. Similarly, for transport, regardless of whether it is simple diffusion along a concentration gradient or any of the variants of active transport, the distance from the site of synthesis (or the volume of the cell during diffusion) is greater in a large cell. Cell volume can also be important in the degradation of proteins during the transition from one functional state of the cell to another.

Comment 9.

l.753. How the AA observed the increase of Sas6?

Answer 9. The details of gene expression analysis were added in Material and Methods section (Lines 168-173)

« 2.4. Gene expression

Transcriptome data on bovine preimplantation embryonic development were retrieved from the NCBI Gene Expression Omnibus database (GEO, https://www.ncbi.nlm.nih.gov/geo/; GEO dataset accession GDS3960). Mean expression values of genes PLK4, SASS6, and CEP192 were calculated from the values reported from two independent microarray hybridizations of mRNA from bovine oocytes, zygotes, 2-cell (2C), 4-cell (4C), 8-cell (8C), 16-cell (16C), morula, and blastocysts. Gene expression values at different stage of embryo development were presented as a histogram. »